

# Fourier-transformed gauge theory models of three-dimensional topological orders with gapped boundaries

Siyuan Wang[1], Yanyan Chen[1], Hongyu Wang[1], Yuting Hu[2⋆] and Yidun Wan[1,3,4†]

**1** State Key Laboratory of Surface Physics, Department of Physics,
Center for Field Theory and Particle Physics, and
Institute for Nanoelectronic devices and Quantum computing,
Fudan University, Shanghai 200433, China
**2** School of Physics, Hangzhou Normal University, Hangzhou 311121, China
**3** Shanghai Research Center for Quantum Sciences, 99 Xiupu Road, Shanghai 201315, China
**4** Hefei National Laboratory, Hefei 230088, China

⋆ yuting.phys@gmail.com , † ydwan@fudan.edu.cn

## Abstract

In this paper, we apply the method of Fourier transform and basis rewriting developed in [1] for the two-dimensional quantum double model of topological orders to the three-dimensional gauge theory model (with a gauge group $G$) of three-dimensional topological orders. We find that the gapped boundary condition of the gauge theory model is characterized by a Frobenius algebra in the representation category $\mathcal{R}ep(G)$ of $G$, which also describes charge splitting and condensation on the boundary. We also show that our Fourier transform maps the three-dimensional gauge theory model with input data $G$ to the Walker-Wang model with input data $\mathcal{R}ep(G)$ on a trivalent lattice with dangling edges, after truncating the Hilbert space by projecting all dangling edges to the trivial representation of $G$. This Fourier transform also provides a systematic construction of the gapped boundary theory of the Walker-Wang model. This establishes a correspondence between two types of topological field theories: the extended Dijkgraaf-Witten and extended Crane-Yetter theories.



## Contents



# 1 Introduction

Exactly solvable lattice models are a useful tool for studying topologically ordered matter phases (topological orders, for short). For instance, the Kitaev quantum double model (QD) [2], the Levin-Wen model (LW) [3], and the twisted quantum double model (TQD) [4] describe two-dimensional topological orders, while the Walker-Wang model (WW) [5] and the twisted gauge theory (TGT) model [6] describe three-dimensional topological orders. In two dimensions, the QD model and the LW model are related by Fourier transform and basis rewriting [1, 7, 8], while the relationship between the TGT model and the WW model is still less understood.

## 1.1 Key results

In this paper, we apply the method of Fourier transform and basis rewriting developed in [1] for the two-dimensional QD model of topological orders to the three-dimensional untwisted gauge theory (GT) model of three-dimensional topological orders. The following are the key results.

- The Fourier-transfomed GT model with input data $G$ and the WW model with input data $\text{Rep}(G)$ are exactly identical by comparing their Hilbert spaces and Hamiltonians. A full set of gapped boundaries, including rough boundaries which has not been explored in the existing literature, can thus be constructed for the WW model with input data $\text{Rep}(G)$ through the Fourier transform method.

- Similar with the two-dimensional cases, the gapped boundaries of the WW model and the Fourier-transfomed GT model are specified by the Frobenius algebra $A_{G,K}$, where

finite groups $G$ and $K$ are the bulk and boundary input data of the original GT model respectively. The Frobenius algebra $A_{G,K}$ explicitly indicates charge condensation and splitting at the boundary.

## 1.2 Backgrounds and motivations

Two-dimensional topological orders have been well studied though the QD model, which is the Hamiltonian extension of the untwisted Dijkgraaf-Witten theory [9], and the LW model. Despite the apparent differences between the QD model and LW model, the two are intimately related: They are dual to each other in the sense that the QD model with a finite gauge group $G$ as its input data can be mapped to the LW model with input data $\text{Rep}(G)$ (the representation category of $G$) via Fourier transform [1,7,8]. Both the QD and LW models have been extended to the case with gapped boundaries, called the extended QD [10–13] and LW models [14–16]. While the gapped boundary of the extended QD model is specified by a subgroup $K \subseteq G$ of the bulk gauge group $G$, that of the extended LW model is specified by a Frobenius algebra object of the input fusion category of the bulk. The extended QD and LW models can also be mapped to each other via Fourier transform and basis rewriting [1]. The boundary input data $K \subseteq G$ of the extended QD model is mapped to the boundary input Frobenius algebra $A_{G,K} \in \text{Obj}(\text{Rep}(G))$. The gapped boundaries of two-dimensional topological orders are well understood through anyon condensation[1] [12, 14, 22–28], where the charge condensation at the boundary is observed from the input data of the boundary via the Fourier transform [1]. This Fourier transform and basis rewriting also manifest the full electromagnetic (EM) duality in these models [1]. It is believed that the understanding of the relationship between the QD model and the LW model is now complete. The EM duality in the TQD model is also studied [28].

In contrast to two dimensions, the models of three-dimensional topological orders and their exact relations are less understood. The TGT model is believed to describe all possible three-dimensional topological orders with point-like bosonic excitations [29,30]. The TGT model is specified by an input finite gauge group $G$ and a 4-cocycle $\omega \in H^4[G, U(1)]$, and the gapped boundary of the TGT model is specified by a subgroup $K$ and a 3-cocycle $\alpha \in H^3[G, U(1)]$ [6]. We will explore charge condensation in the untwisted TGT model, i.e., the GT model, which reveals the relationship between the bulk and the gapped boundary of the GT model. In two dimensions, the bulk charges may split and partially or fully condense at the boundary, depending on the gapped boundary condition [1]. In three dimensions, however, the phenomenon of boundary charge condensation has not been fully understood from the perspective of the lattice model [31]. Although layer construction offers another approach to understanding charge condensation at the boundary of certain three-dimensional topological orders [32,33], most current understandings of this phenomenon are either categorical and abstract or limited to specific cases [31,34]. Therefore, investigating the splitting and partial condensation phenomena in concrete lattice models of three-dimensional topological orders with general input data would be worthwhile. In this paper, we generalize the method of Fourier transform and basis rewriting developed in [1] from two to three dimensions to investigate the gapped boundaries of a three-dimensional GT model. We prove that the boundary theory of the Fourier-transformed model is also characterized by the Frobenius algebra. Then, similar to the two-dimensional extended LW model with input data $\text{Rep}(G)$, the charge of the condensates is described by the Frobenius algebra and the splitting process originates from the multiplicity of the objects of $\text{Rep}(G)$. Thus, we explain these phenomena using only the input data of the three-dimensional model.

---

[1]The underlying mathematics of anyon condensation was firstly presented in RCFT [17]. Then, the concept of anyon condensation was firstly introduced in the context of $(2 + 1)$-dimensional TQFT [18, 19]. The first works discussing anyon condensation in lattice models were [20, 21].

On the other hand, the three-dimensional WW model is specified by an input unitary braided fusion category (UBFC) and describes a family of three-dimensional topological orders [35, 36]. In this paper, we show that our Fourier transform, extending the mapping process from QD models to LW models into three dimensions, indeed maps a three-dimensional GT model with input data $G$ to a WW model with UBFC Rep($G$) as its input data on the level of Hilbert spaces and Hamiltonians.[2] Furthermore, the Fourier transform of the gapped boundaries of the three-dimensional GT model also gives a full systematic construction of the gapped boundaries of the WW model with UBFC Rep($G$) as input data. Here the phrase "UBFC Rep($G$)" means the representation category of finite group $G$ equipped with a braiding structure. Prior to our work, a canonical smooth boundary was described explicitly in three-dimensional toric code and double-semion, with a detailed discussion of the resulting boundary excitations [38], and a second class of gapped boundary conditions for WW models was discussed in detail and in full generalty [39]. Nevertheless, no systematic construction of the rough boundary Hamiltonian of the WW model has been established. Our understanding of the relationship between the TGT model and the WW model is also limited to some special cases, such as the example of the $\mathbb{Z}_2 \times \mathbb{Z}_2$ twisted gauge theory model, which is shown to share the same modular matrices with the corresponding WW model [36]

In this paper, we focus on three-dimensional GT models defined on cubic lattices with boundaries, with input data being a finite group $G$ in the bulk and a subgroup $K \subseteq G$ within the boundary. The more complicated cases, i.e., the twisted gauge theory models and twisted boundaries, are our ongoing work and will be reported elsewhere. The Fourier-transformed basis of the Hilbert space of the GT model leads us to rewrite the model on a slightly different trivalent lattice $\tilde{\Gamma}$ with a tail (dangling edge) attached to each vertex. This lattice is precisely where the WW model lives on with an enlarged Hilbert space. This enlargement is necessary since the original WW model has a Hilbert space insufficient for accommodating the full spectrum of charge excitation, which is analogous to the extended LW model [40] where tails are added to enlarge the orginal Hilbert space of the LW model. After the Fourier transform, the bulk input data becomes a UBFC Rep($G$), while the boundary degrees of freedom are projected into Frobenius algebra $A_{G,K}$, as in the case of Fourier-transformed QD model in two dimensions. We also show that the Fourier-transformed GT model with input data $G$ on the revised lattice $\tilde{\Gamma}$ can be mapped to a WW model with input data Rep($G$) on the same lattice after truncating the Hilbert space by projecting all dangling edges to trivial representation. Since both GT and WW models with gapped boundaries serve as Hamiltonian extensions of the extended untwisted Dijkgraaf-Witten and extended Crane-Yetter topological field theories, our results also establish a correspondence between these two types of topological field theories.

Our paper is organized as follows. Section 2 reviews the three-dimensional GT model with gapped boundaries. Section 3 Fourier transforms and rewrites the extended GT model. Section 4 verifies the emergent Frobenius algebra structure on the boundary. Section 5 proves that our Fourier transform indeed maps the three-dimensional GT model to the WW model. Finally, the appendices collect a review of the WW model and certain details to avoid clutter in the main text.

## 2 Three-dimensional GT model with gapped boundaries

The GT model with gapped boundaries is a Hamiltonian extension of the Dijkgraaf-Witten topological gauge theory with a finite gauge group. Without taking twist into consideration,

---

[2]The Fourier transform discussed in this paper is similar to the transformation between the electric and magnetic bases of the non-Abelian lattice gauge theory firstly introduced by Kogut [37]. For a concrete comparison, see Section 3.2.

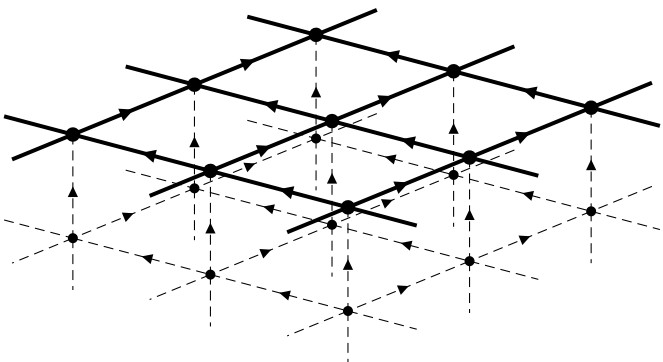

Figure 1: A part of an oriented cubic lattice, on which the three-dimensional GT model with gapped boundaries is defined. Each edge of the lattice is assigned with a group element of a finite gauge group $G$. The thick lines comprise the boundary while the dashed lines comprise the bulk.

the model can be defined on an arbitrary lattice with one or multiple boundaries. Topological invariance enables us to define the model on a specific lattice for simplicity. In this paper, we consider an oriented cubic lattice $\Gamma$ with boundaries, part of which is shown in Figure 1.

The input data of the bulk theory is a finite gauge group $G$. The total Hilbert space is the tensor product of the local Hilbert spaces of each edge of $\Gamma$, which is spanned by the basis $\{|g\rangle\}_{g \in G}$. The total Hilbert space is thus given by

$$\mathcal{H}_G^{\text{GT}} = \bigotimes_{e \in \Gamma} \mathcal{H}_e = \bigotimes_{e \in \Gamma} \text{span}_{g_e \in G}\{|g_e\rangle\}, \tag{1}$$

where $e$ is an edge in $\Gamma$. The local basis vector $|g_e\rangle$ is invariant if we reverse the direction of the edge $e$ and take the inverse of $g_e$ as $\bar{g}_e = g_e^{-1}$ simultaneously. In this paper, we will work with the fixed orientation shown in Figure 1. The inner product in the local Hilbert space is simply $\langle g_e'|g_e\rangle = \delta_{g'g}$. The Hamiltonian of the model is the sum of a bulk Hamiltonian and a boundary Hamiltonian:

$$H_{G,K}^{\text{GT}} = H_G^{\text{GT}} + \overline{H_K^{\text{GT}}}. \tag{2}$$

Here and hereafter, an operator with an overline is a boundary operator. The bulk Hamiltonian consists of the sum of vertex operators and that of plaquette operators:

$$H_G^{\text{GT}} = -\sum_{v \in \Gamma \setminus \partial \Gamma} A_v^{\text{GT}} - \sum_{p \in \Gamma \setminus \partial \Gamma} B_p^{\text{GT}}, \tag{3}$$

where the sums run over all vertices and plaquettes in the bulk of $\Gamma$. The vertex operator $A_v^{\text{GT}}$ acts locally on the six-valent vertex $v$:

$$A_v^{\text{GT}}\left| \begin{array}{c} \includegraphics \end{array} \right\rangle = \frac{1}{|G|}\sum_{x \in G} A_v^x \left| \begin{array}{c} \includegraphics \end{array} \right\rangle = \frac{1}{|G|}\sum_{x \in G} \left| \begin{array}{c} \includegraphics \end{array} \right\rangle, \tag{4}$$

where $A_v^x$ is a local discrete gauge transformation given by the group element $x \in G$, and thus $A_v$ is the gauge transformation averaged over $G$. Clearly, $A_v^{\text{GT}}$ is a projector which projects out local states that are not invariant under the gauge transformation. The plaquette operator acts

locally on the four edges bounding the plaquette $p$:

$$B_p^{\text{GT}}\left| \begin{array}{c} g \quad j \\ p \\ h \quad i \end{array} \right\rangle = \delta_{g\bar{j}\bar{i}h,e}\left| \begin{array}{c} g \quad j \\ p \\ h \quad i \end{array} \right\rangle, \tag{5}$$

which is also a projector that projects out local states with non-vanishing flux through the plaquette $p$. All plaquette operators and vertex operators commute.

The gapped boundary condition of the three-dimensional GT model is characterized by a subgroup $K \subseteq G$ and a 3-cocycle $\alpha \in H^3[G, U(1)]$. In this paper, we will focus on boundaries with the trivial $\alpha$, i.e. $\alpha \equiv 1$. The boundary Hamiltonian consists of respectively the sums of boundary vertex, plaquette, and edge operators:

$$\overline{H_K^{\text{GT}}} = -\sum_{v \in \partial\Gamma} \overline{A_v^{\text{GT}}} - \sum_{p \in \partial\Gamma} \overline{B_p^{\text{GT}}} - \sum_{e \in \partial\Gamma} \overline{C_e^{\text{GT}}}, \tag{6}$$

where the sums run over all the boundary vertices, plaquettes, and edges on the boundary $\partial\Gamma$ of $\Gamma$. The definition of the boundary vertex operators and plaquette operators are similar to the bulk operators: $\overline{A_v^{\text{GT}}}$ is again defined as a gauge transformation averaged instead in the subgroup $K$, and the definition of $\overline{B_p^{\text{GT}}}$ is just the same as $B_p^{\text{GT}}$. The edge operator $\overline{C_e^{\text{GT}}}$ is a projector:

$$\overline{C_e^{\text{GT}}}\left| \begin{array}{c} l \\ e \end{array} \right\rangle = \delta_{l \in K}\left| \begin{array}{c} l \\ e \end{array} \right\rangle, \tag{7}$$

which projects the boundary degrees of freedom into the subgroup $K$. The boundary vertex, plaquette, and edge operators all commute with each other and with the bulk vertex and plaquette operators. Therefore, the total Hamiltonian (2) is exactly solvable. The ground states are the common +1 eigenstates of all operators in the total Hamiltonian. The ground state degeneracy (GSD) can be computed by

$$\text{GSD} = \text{Tr}\left( \prod_{v \in \Gamma \backslash \partial\Gamma} A_v^{\text{GT}} \prod_{p \in \Gamma \backslash \partial\Gamma} B_p^{\text{GT}} \prod_{v \in \partial\Gamma} \overline{A_v^{\text{GT}}} \prod_{p \in \partial\Gamma} \overline{B_p^{\text{GT}}} \prod_{e \in \partial\Gamma} \overline{C_e^{\text{GT}}} \right), \tag{8}$$

where the trace is taken over the total Hilbert space (1). The elementary excitations in the model without boundary are charges on the vertices and loop-like excitations. A charge at vertex $v$ arises when $A_v = 0$; a loop-like excitation occurs when $B_p = 0$ on a series of plaquettes which forms a loop [41, 42]. If the gapped boundary is included, there still exists one more type of elementary excitations, that is, the bulk string-like excitations that terminate at gapped boundaries (see Figure 2). In three dimensions, the braiding between point-like charges is trivial, while the braiding between loop-like excitations or string-like excitations is highly non-trivial. Thus, despite some recent progress [29, 42–45], our understanding of loop-like excitations in three-dimensional topological orders is still incomplete.

## 3 Fourier transform the three-dimensional GT model with gapped boundaries

In this section, we Fourier-transform the basis of the Hilbert space of the three-dimensional GT model with gapped boundaries from the group space to the representation space, which allows us to rewrite the three-dimensional GT model with gapped boundaries on a trivalent lattice.

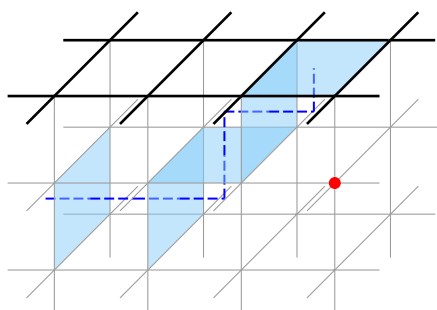

Figure 2: Charge excitation (red dot) and string-like excitation (deep blue dashed line, consisting of a series of light blue plaquettes where the local flatness condition is violated) that terminates on the boundary in the three-dimensional GT model with gapped boundaries.

We then Fourier-transform the boundary Hamiltonian of the three-dimensional GT model. The transformation given by (32), (33) and (39), is determined by the data $\{C_{\mu\nu\rho}, R^{\rho}_{\mu\nu}\}$, where $C_{\mu\nu\rho}$ is the $3j$-symbol of the irreducible representations of the input finite group $G$, and $R^{\rho}_{\mu\nu}$ is the $R$-matrix of the representation category Rep($G$). Generally, given the input finite group $G$, the data $\{C_{\mu\nu\rho}, R^{\rho}_{\mu\nu}\}$ cannot be uniquely determined, which gives a series of transformations. In the following subsection, we will introduce a convention to uniquely determine the $3j$-symbol, while the discussion of the $R$-matrix will be left to subsection 3.2.

## 3.1 A graphical tool for group representation theory

To make derivations easier in this paper, especially the Fourier transform, we employ the subsequent graphical tools for group representation theory as put forth in [46]. Denote by $L_G$ the collection of every unitary irreducible representation $V_\mu$ (up to equivalence) of a finite group denoted by $G$. We will use Greek indices $\mu$ to label irreducible representations and Latin indices $m_\mu$ to label the basis of the representation space $V_\mu$.

**Duality map**   The duality map $\omega_\mu$ is an intertwiner

$$\omega_\mu : \; \mathbb{C} \to V_\mu \otimes V_\mu, \quad 1 \mapsto \sum_{m_\mu, n_{\mu^*}} \Omega^{\mu}_{m_\mu n_{\mu^*}} e_{m_\mu} \otimes e_{n_{\mu^*}}, \tag{9}$$

where $\mu^* \in L_G$ is the dual of the irreducible representation $\mu \in L_G$. The dual representation $\mu^*$ is equivalent (but not necessarily identical) to the complex conjugate representation of $\mu$. The intertwining property of $\omega_\mu$ implies that the complex matrix $\Omega^{\mu}_{m_\mu n_{\mu^*}}$ maps $\mu$ to $\mu^*$ by similarity transformation

$$(\Omega^\mu)^{-1} D^\mu(g) \Omega^\mu = (D^{\mu^*}(g))^*, \tag{10}$$

where $D^\mu(g)$ is the representation matrix of group element $g$. We also require that $\Omega^{\mu}_{m_\mu n_{\mu^*}}$ satisfies the normalization condition $\Omega^\dagger \Omega = \mathbb{1}$.

Within the normalization condition, the matrix $\Omega^\mu$ can only be determined up to a phase factor. If $\mu$ is self-dual, then the transpose of $\Omega^\mu$ is also a duality map, which implies that $\Omega^\mu = \beta_\mu (\Omega^\mu)^{\mathrm{T}}$ with $\beta_\mu = \pm 1$. It can be proved that $\beta_\mu$ is completely determined by $\mu$ and thus can be viewed as an intrinsic property of $\mu$, called the Forbenius-Schur (FS) indicator. Moreover, if $\mu$ is not self-dual, we always set $\beta_\mu = 1$.

Graphically, the duality map and its inverse have presentations

$$
\equiv \Omega^{\mu}_{m_{\mu} n_{\mu^*}}\,, \qquad\qquad \equiv [(\Omega^{\mu})^{-1}]_{m_{\mu^*} n_{\mu}}\,. \tag{11}
$$

Being an intertwiner, the duality map commutes with group actions, namely

$$
= \,, \tag{12}
$$

where we have introduced the graphical presentation for group actions:

$$
= D^{\mu}_{m_{\mu} n_{\mu}}(g)\,, \tag{13}
$$

and the contraction of Latin indices is presented by concatenating two lines. Note that the direction of all lines in our graphical presentation are upward by default. A line with label $\mu$ directed downward should always be regarded as the line with label $\mu^*$ directed upward.

Since all irreducible representations $\mu \in L_G$ are unitary,

$$
[D^{\mu}_{m_{\mu} n_{\mu}}(g)]^* = D^{\mu}_{n_{\mu} m_{\mu}}(\bar{g}) \;\Rightarrow\; \left[\;\right]^* = \;. \tag{14}
$$

Then, (10) can be graphically presented as

$$
= \;. \tag{15}
$$

**3$j$-symbol**  Frequently in later derivations, we will need 3$j$-symbols to deal with the coupling of three representations of $G$. A 3$j$-symbol is a tensor $C_{\mu\nu\rho;m_{\mu}m_{\nu}m_{\rho}}$ that is defined as an intertwiner:

$$
C_{\mu\nu\rho} : \; V_{\mu} \otimes V_{\nu} \otimes V_{\rho} \to \mathbb{C}\,, \quad |\mu m_{\mu}, \nu m_{\nu}, \rho m_{\rho}\rangle \mapsto C_{\mu\nu\rho;m_{\mu}m_{\nu}m_{\rho}}\,. \tag{16}
$$

A 3$j$-symbol is depicted as

$$
\equiv C_{\mu\nu\rho;m_{\mu}m_{\nu}m_{\rho}}\,, \qquad\qquad \equiv (C_{\mu\nu\rho;m_{\mu}m_{\nu}m_{\rho}})^*\,. \tag{17}
$$

The normalization condition of $3j$-symbols can then be presented as

$$\sum_{m_\mu m_\nu m_\rho} C_{\mu\nu\rho;m_\mu m_\nu m_\rho}(C_{\mu\nu\rho;m_\mu m_\nu m_\rho})^* = \mu \bigcirc \nu \bigcirc \rho = 1. \tag{18}$$

Being an intertwiner, a $3j$-symbol is invariant under group actions, namely

$$= \tag{19}$$

In this paper, we will also use a lot of Clebsch-Gordan (CG) coefficients $C^\rho_{\mu\nu} : V_\mu \otimes V_\nu \to V_\rho$ and $C^{\mu\nu}_\rho : V_\rho \to V_\mu \otimes V_\nu$, which can be defined using $3j$-symbols and duality maps:

$$C^{\rho^*m_{\rho^*}}_{\mu\nu;m_\mu m_\nu} \equiv \quad\equiv\quad = \sum_{m_\rho} C_{\mu\nu\rho;m_\mu m_\nu m_\rho} \Omega^\rho_{m_\rho m_{\rho^*}}, \tag{20}$$

$$C^{\mu\nu;m_\mu m_\nu}_{\rho^*m_{\rho^*}} \equiv \quad\equiv\quad . \tag{21}$$

The CG coefficients enable the following basis transformation:

$$|\mu m_\mu\rangle \otimes |\nu m_\nu\rangle = \sum_{\rho,m_\rho} C^{\mu\nu;m_\mu m_\nu}_{\rho m_\rho} |\rho m_\rho\rangle, \tag{22}$$

which is the foundation of rewriting the basis of the Fourier-transformed Hilbert space on a trivalent lattice.

For later convenience, we list a few properties of $3j$-symbols as follows. For a generic group $G$, we can always construct such $3j$-symbols satisfying the following properties [46]:

$$= \frac{1}{|G|} \sum_{g\in G} \quad = \quad , \tag{23}$$

where $\Gamma_{\text{closed}}$ means that the part of the graph does not have any open edges. Moreover, we have the following orthogonality conditions:

$$\sum_\rho \tilde{d}_\rho \quad = \delta_{m_\mu m'_\mu} \delta_{m_\nu m'_\nu}, \qquad = \frac{1}{\tilde{d}_\rho} \delta_{\rho'\rho} \delta_{m'_\rho m_\rho}, \tag{24}$$

where $\tilde{d}_\mu = \beta_\mu d_\mu$ is the quantum dimension of the representation $\mu$.

Similarly to duality maps, a 3$j$-symbol $C_{\mu\nu\rho}$ is also determined up to a complex number. In order to construct symmetrized 6$j$-symbols via 3$j$-symbols, we further require that

$$C_{\mu\nu\rho;m_\mu m_\nu m_\rho} = \beta_\rho C_{\rho\mu\nu;m_\rho m_\mu m_\nu}, \tag{25}$$

and that

$$(C_{\mu\nu\rho;m_\mu m_\nu m_\rho})^* = \sum_{m_{\mu^*} m_{\nu^*} m_{\rho^*}} C_{\rho^*\nu^*\mu^*;m_{\rho^*} m_{\nu^*} m_{\mu^*}} \Omega^{\rho^*}_{m_{\rho^*} m_\rho} \Omega^{\nu^*}_{m_{\nu^*} m_\nu} \Omega^{\mu^*}_{m_{\mu^*} m_\mu}. \tag{26}$$

Combining the two requirements above and the definition of 3$j$-symbols, we can completely determine all 3$j$-symbols for a given finite group $G$. Note that (25) yields $\beta_\mu \beta_\nu \beta_\rho = 1$ if $C_{\mu\nu\rho}$ does not vanish. Some examples are listed in [46].

**Symmetrized 6$j$-symbol**    A 6$j$-symbol $F : L_G^6 \to \mathbb{C}$ is defined by

$$\text{(figure)} = \sum_\rho F^{\mu\nu\lambda}_{\eta\kappa\rho} \text{ (figure)}, \tag{27}$$

where we have omitted Latin indices $m_\mu, m_\nu, \ldots$. The above expression relates the two equivalent ways of decomposing the tensor product representation $V_{\nu^*} \otimes V_{\mu^*} \otimes V_{\kappa^*}$. Composing both sides of the above equation by the duality maps and the 3$j$-symbols in an appropriate way as:

$$\text{(figure)} = \sum_{\rho'} F^{\mu\nu\lambda}_{\eta\kappa\rho'} \text{ (figure)},$$

which by the second equation of (24) becomes

$$F^{\mu\nu\lambda}_{\eta\kappa\rho} = \tilde{d}_\rho \text{ (figure)} \equiv \tilde{d}_\rho G^{\mu\nu\lambda}_{\eta\kappa\rho}. \tag{28}$$

Here we introduce the symmetrized 6$j$-symbols, given by $G^{\mu\nu\lambda}_{\eta\kappa\rho} = F^{\mu\nu\lambda}_{\eta\kappa\rho}/\tilde{d}_\rho$. In terms of 3$j$-symbols and duality maps, we have

$$\begin{aligned}
G^{\mu\nu\lambda}_{\eta\kappa\rho} = \sum_{\text{all } m\text{'s and } n\text{'s}} &\Omega^\mu_{m_\mu n_{\mu^*}} \Omega^\nu_{m_\nu n_{\nu^*}} \Omega^\lambda_{m_\lambda n_{\lambda^*}} \Omega^\eta_{m_\eta n_{\eta^*}} \Omega^\kappa_{m_\kappa n_{\kappa^*}} \Omega^\rho_{m_\rho n_{\rho^*}} \\
&\times C^{\kappa\lambda^*\eta}_{m_\kappa n_{\lambda^*} m_\eta} C^{\eta^*\nu^*\rho}_{n_{\eta^*} n_{\nu^*} m_\rho} C^{\rho^*\mu^*\kappa^*}_{n_{\rho^*} n_{\mu^*} n_{\kappa^*}} C^{\mu\nu\lambda}_{m_\mu m_\nu m_\lambda}.
\end{aligned} \tag{29}$$

Graphically, symmetrized $6j$-symbols can be presented either by a planar graph or a tetrahedron

$$G^{\mu\nu\lambda}_{\eta\kappa\rho} = \eta^* \quad \cdots \quad = \eta^* \quad \cdots \quad \equiv \eta \quad \cdots \quad . \tag{30}$$

The tetrahedral symmetry of $G^{\mu\nu\lambda}_{\eta\kappa\rho}$ then follows from (30).

**Other conventions** The great orthogonality theorem of finite group representations will be often used in our paper; it can be presented graphically as

$$\frac{1}{|G|}\sum_{g\in G} \quad \cdots \quad = \frac{1}{d_\mu}\delta_{\mu\nu}\delta_{m_\mu n_\nu}\delta_{n_\mu m_\nu}, \tag{31}$$

which yields

$$\frac{1}{|G|}\sum_{g\in G} \quad \cdots \quad = \frac{1}{d_\mu}\delta_{\mu\nu} \quad \cdots \quad .$$

Here and hereafter, the horizontal lines should be understood through the following convention:

$$\cdots \quad \equiv \quad \cdots \quad , \qquad \cdots \quad \equiv \quad \cdots \quad .$$

### 3.2 Fourier transform on the Hilbert space

The Fourier transform of the operators in the Hamiltonian requires defining the Fourier transform of the Hilbert space of a three-dimensional GT model with gapped boundaries with $G$ as input data. In this subsection, we will first discuss the Fourier transform on the local Hilbert space of a vertex. Then, we will show that to define the Fourier transform on the total Hilbert space, the braiding structure must be introduced.

**A sketch of the Fourier transform on the Hilbert space** In this subsection, we describe this Fourier transform step by step. The explicit transformation will be dealt with a little later. Our focus here lies in the logic and physics of the basis transformations.

*Step 1: Fourier transform the local Hilbert space*

The total Hilbert space $\mathcal{H}$ of the model is the tensor product of all local Hilbert spaces $\mathcal{H}_e$ on the edges. The basis vector $|g\rangle$ of $\mathcal{H}_e$ Fourier-transforms as:

$$|\mu, m_\mu, n_\mu\rangle = \frac{\nu_\mu}{\sqrt{G}}\sum_{g\in G} D^\mu_{m_\mu n_\mu}(g)|g\rangle, \tag{32}$$

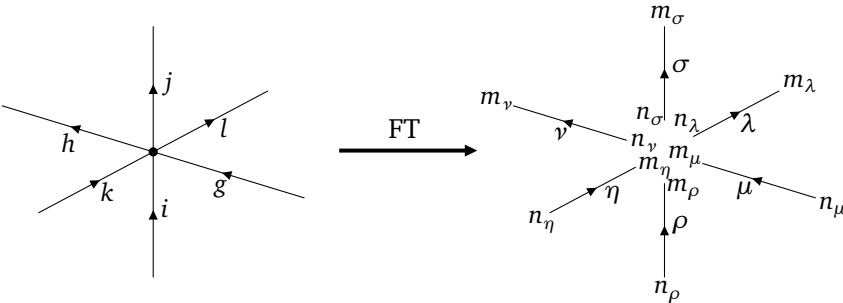

Figure 3: The Fourier transform of a six-valent vertex in the bulk of the lattice $\Gamma$ on which the three-dimensional GT model is defined.

where $v_\mu = d_\mu^{1/2}$. The basis transformation (32) is exactly the finite group version of the transformation between the electric and magnetic bases of non-Abelian lattice gauge theory:

$$|j, m, m'\rangle = \int_{SU(2)} dU \sqrt{2j+1} D_{mm'}^j(U)|U\rangle,$$

where $D_{mm'}^j(U)$ is the Wigner $D$-matrix, and $dU$ is the Haar measure on the SU(2) group manifold. We dub the Fourier-transformed basis $\{|\mu, m_\mu, n_\mu\rangle\}$ the local rep-basis. The local rep-basis $\{|\mu, m_\mu, n_\mu\rangle\}$ and the group-basis $\{|g\rangle\}$ have the same dimension due to $\sum_{\mu \in L_G} d_\mu^2 = |G|$. Furthermore, using the great orthogonality theorem, we can prove that the rep-basis is also orthonormal: $\langle \mu', m', n'|\mu, m, n\rangle = \delta_{\mu'\mu}\delta_{m'm}\delta_{n'n}$. We can Fourier transform $\mathcal{H}_e$ on each individual edge independently, resulting in a linear superposition of the rep-basis states, shown in Figure 3. Here, the orientations of the rep-basis states inherit from those of the group-basis states.

*Step 2: Rewrite the rep-basis to get a trivalent lattice*

Although the rep-basis is orthonormal and has the same dimension as the group basis $\{|g\rangle\}$, it cannot describe the states of a lattice model because it does not contain vertices. Therefore, we need to rewrite the rep-basis state and identify it as a trivalent lattice.

Recall the definition of CG coefficients (22), which can be graphically presented as a basis rewriting:

$$\left| \begin{matrix} m_\mu & m_\nu \\ \mu & \nu \\ n_\mu & n_\nu \end{matrix} \right\rangle = \sum_{\lambda, m_\lambda} v_\lambda \left( \begin{matrix} m_\mu & m_\nu \\ \mu & \nu \\ \lambda \\ m_\lambda \end{matrix} \right) \left| \begin{matrix} m_\lambda \\ \lambda \\ \mu & \nu \\ n_\mu & n_\nu \end{matrix} \right\rangle. \tag{33}$$

The orthonormality of the rewritten basis still comes from the normalization condition of the $3j$-symbol. The rewritten basis also has the same dimension as the rep-basis due to the defining property of the $3j$-symbol. By the equation above, we can fuse the six representations and six Latin indices in a specific order at the vertex, as depicted in Figure 4.

*Step 3: Add a dangling edge*

As shown in Figure 4, we first fuse $\mu$ and $\lambda$ by contracting there indices $m_\mu$ and $n_\lambda$, resulting in a linear combination of irreducible representations $\{\alpha\}$ with a free end labeled $m_\alpha$. Then, we fuse $\alpha$ and $\sigma$ by contracting their indices $m_\alpha$ and $n_\sigma$, which results in another linear combination of irreducible representations $\{\beta\}$ with another free end. Repeating this procedure and, in the end, we need to fuse $\nu$ and $\delta$. As $\nu$ and $\delta$ are not always the same, in order to

Figure 4: Rewrite the rep-basis as defined on a trivalent lattice. In the first step, we fuse $\mu$ and $\lambda$ by contracting their indices $m_\mu$ and $n_\lambda$, which results in a linear combination of irreducible representations $\{\alpha\}$ with a free end and labeled by $m_\alpha$. Repeating this procedure and in the end, we fuse $\delta$ and $\nu$ and obtain a tail attached to the original vertex with an free end, labeled by $(s, m_s)$.

maintain the correct number of local degrees of freedom, we need to add an extra dangling edge $s$ with a free end labeled by $m_s$, as shown in the final result in Figure 4. Such a dangling edge will be labeled by $\{s, m_s\}$ and called a tail for short.

So far we have rewritten the basis of the total Hilbert space as defined on an actual trivalent lattice $\tilde{\Gamma}$, with a tail attached to each of the original vertices. Note that these tails do not belong to any plaquette in $\tilde{\Gamma}$. If we restrict the total Hilbert space to a subspace in which the degrees of freedom on the tails are all projected to the trivial representation, the newly obtained lattice $\tilde{\Gamma}$ can serve as a suitable lattice for defining a WW model. We will discuss the relationship between our Fourier-transformed GT model and the WW model in section 5 and revisit this point therein.

**The Fourier transform of a boundary 5-valent vertex** To better understand the Fourier transform and basis rewriting illustrated in Figure 3 and Figure 4, let us consider the local Hilbert space of a boundary 5-valent vertex in detail. By doing so, we aim to obtain a precise linear transformation of the basis with commensurate coefficients. In our graphical representation and by (32), the Fourier transform of the group-basis to the rep-basis of the local Hilbert space reads

$$
\left| \begin{array}{c} \text{(diagram)} \end{array} \right\rangle = \frac{\nu_\mu \nu_\nu \nu_\rho \nu_\eta \nu_\lambda}{\sqrt{|G|^5}} \sum_{jikgh \in G} \left( \begin{array}{c} \text{(diagram)} \end{array} \right) \left| \begin{array}{c} \text{(diagram)} \end{array} \right\rangle . \tag{34}
$$

Here, the edges $i$, $j$, $g$, and $h$ are lying on the boundary, while the edge $k$ lies in the bulk. We denote the group-basis and rep-basis states in the above equation as $|jikgh\rangle$ and $|\mu\nu\rho\eta\lambda\rangle$. Then, we can rewrite the rep-basis by first fusing $\mu$ and $\rho$, resulting in a set of representations $\{\alpha\}$, and then fuse $\alpha$ and $\lambda$ to get $\beta$, and then fuse $\beta$ and $\eta$ to get $\gamma$, and finally we fuse $\gamma$ and

$\nu$, resulting in a dangling edge $\{s, m_s\}$. Using (33), this procedure yields four $3j$-symbols with some indices contracted, yielding the expansion coefficients:

$$|\mu\nu\rho\eta\lambda\rangle = \sum_{\alpha\beta\gamma\in L_G}\sum_{s,m_s} v_\alpha v_\beta v_\gamma v_s \left( \text{[diagram]} \middle| \text{[diagram]} \right),$$

(35)

where the coefficients $v_\alpha v_\beta v_\gamma v_s$ are introduced such that the rewritten basis is still orthonormal. We denote the rewritten rep-basis state at the vertex by $|\Psi_{sm_s}\rangle$ and write down the inverse transformation as:

$$|\Psi_{sm_s}\rangle = \sum_{\substack{m_\mu m_\rho m_\eta \\ n_\lambda n_\nu m_s}} v_\alpha v_\beta v_\gamma v_s \left( \text{[diagram]} \right) |\mu\nu\rho\eta\lambda\rangle.$$

(36)

The linear transformation (35) rewrites the subspace spanned by $\{m_\mu, m_\rho, m_\eta, n_\nu, n_\lambda\}$ of the local Hilbert space spanned by $\{\mu\nu\rho\eta\lambda\}$. Under this transformation, the degrees of freedom $m_\mu$, $m_\rho$, $m_\eta$, $n_\nu$, and $n_\lambda$, which are all independent of each other, are transformed into $\alpha, \beta, \gamma, s$, and $m_s$, which are not all independent, while other degrees of freedom remain unchanged. In fact, as the $3j$-symbol ensures an intertwiner space at each vertex, $\alpha, \beta$, and $\gamma$ are determined by the choice of all the irreducible representations labeling those edges with an open end in the local Hilbert space spanned by $|\Psi_{sm_s}\rangle$, i.e., $\mu, \nu, \rho, \eta, \lambda$, and $s$. Thus, we may simplify by labeling the new transformed basis as $|\Psi_{sm_s}\rangle$, while keeping all the other labels in the graph unobvious, which causes no confusion because in the actual calculation, e.g., in computing the inner product of two such local basis states, i.e., $\langle\Psi'_{s'm'_s}|\Psi_{sm_s}\rangle$, the prime in $\Psi'$ indicates that the hidden labels in $\Psi$ should all be primed as well. In the next subsection, we will see another advantage of this simplified notation.

Equations (36) and (34) then lead to the following inner product:

$$\langle jikgh|\Psi_{sm_s}\rangle = \frac{v_\mu v_\nu v_\rho v_\eta v_\lambda v_\alpha v_\beta v_\gamma v_s}{\sqrt{|G|^5}} \left( \text{[diagram]} \right),$$

(37)

which directly defines a transformation between the rewritten rep-basis and the group-basis at a boundary vertex. We prove in Appendix B that the local basis states $|\Psi_{sm_s}\rangle$ are orthonormal and complete so that the inner product defined above is indeed a well-defined basis transformation of the local Hilbert space.

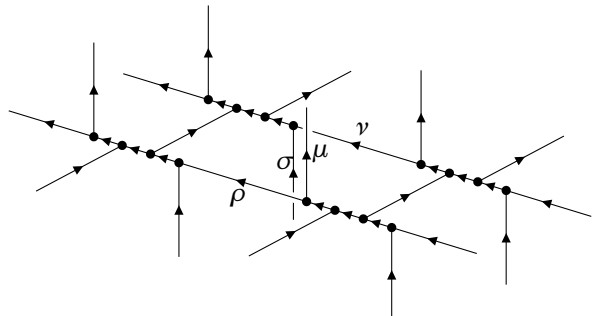

Figure 5: A whole plaquette in the bulk, where the edges labeled by $\mu$ and $\nu$ are crossing with each other, as are edges labeled by $\rho$ and $\sigma$.

**The Fourier transform when edges cross with each other**   So far, our description of the Fourier transform on the Hilbert space is still incomplete, because the total Hilbert space of the Fourier-transformed three-dimensional GT model cannot be trivially viewed as the tensor product of the Fourier-transformed local Hilbert spaces of single vertices defined above. In fact, as will be explained in the following context, a braiding structure must be added.

Without loss of generality, let us consider the state in the Hilbert space with a whole plaquette in the bulk, as shown in Figure 5. One can observe that when a larger Hilbert space is considered, edges unavoidably cross with each other in the rewritten rep-basis. Therefore, defining the inner product between the states that include crossing edges in the rewritten rep-basis and those in the group-basis becomes necessary. To achieve this, let us consider the following inner product.

$$
\left\langle \begin{array}{c} \uparrow h \\ \downarrow g \end{array} \middle| \begin{array}{c} m_\nu \\ m_{\bar\mu} \uparrow \mu n_\mu \\ n_\nu \end{array} \right\rangle \equiv \frac{\nu_\mu \nu_\nu}{|G|} \begin{array}{c} m_\nu \\ \textcircled{h} \\ m_\mu - \textcircled{g} \, \bar\mu \, n_\mu \\ n_\nu \end{array} \overset{?}{=} \frac{\nu_\mu \nu_\nu}{|G|} D^\mu_{m_\mu n_\mu}(g) D^\nu_{m_\nu n_\nu}(h), \tag{38}
$$

where the first equivalence comes from the Fourier-transformation of the basis vector of $\mathcal{H}_e$ (32), and the equality with a question mark "$\overset{?}{=}$" is the evaluation of the graphical presentation if edge crossing is treated trivially.

Clearly, the convention given by "$\overset{?}{=}$" is insufficient because it cannot distinguish the over- and under-crossing relation between the two edges. Nonetheless, we can indeed find a reasonable convention to evaluate the graphical presentation in (38) that encodes the crossing information of the edges in the inner product by introducing a braiding structure. Using the $F$-move given by the first equation in (24), formally we have:

$$
\begin{array}{c} m_\nu \\ \textcircled{h} \\ m_\mu - \textcircled{g} \, \bar\mu \, n_\mu \\ n_\nu \end{array} = \sum_\rho \tilde{d}_\rho \; \begin{array}{c} m_\nu \\ \textcircled{h} \\ m_{\bar\mu} - \textcircled{g} \, \mu \\ \rho \, \mu \, n_\mu \\ \nu \\ n_\nu \end{array} \tag{39}
$$

$$
\equiv \sum_{\rho, m_\rho m'_\mu m'_\nu} \tilde{d}_\rho D^\mu_{m_\mu m'_\mu}(g) D^\nu_{m_\nu m'_\nu}(h) \overline{R^{\mu\nu}_\rho} C^{\mu\nu; m'_\mu m'_\nu}_{\rho m_\rho} C^{\rho m_\rho}_{\nu\mu; n_\nu n_\mu},
$$

where we have introduced new equivalences in the second row above. Namely

$$
\equiv R_\rho^{\mu\nu} \qquad = R_\rho^{\mu\nu} C_{\rho m_\rho}^{\mu\nu;m_\mu m_\nu}, \qquad \equiv \overline{R_\rho^{\mu\nu}} \qquad . \tag{40}
$$

Therefore, during the Fourier transformation, we first fix a projection of the 3D lattice onto 2D, and that any edges that cross in this projection will pick up an $R$-matrix when evaluating the inner product between the Fourier-transformed basis and the group basis.

Here, $R : L_G^3 \to \mathbb{C}$ is called an $R$-matrix, which cannot be expressed explicitly in terms of the duality maps and $3j$-symbols of the representations of a finite group $G$. To determine the $R$-matrix, we impose the following symmetry constraint of the rewritten rep-basis (where Latin indices $m_\mu, m_\nu, \dots$ are omitted):

$$
\left| \vphantom{\Big|} \right\rangle = \left| \vphantom{\Big|} \right\rangle. \tag{41}
$$

As a result, the $R$-matrix elements must be a complex phase and satisfy the Hexagon identity [5]. Moreover, as the topological spins of simple objects in Rep($G$) are always 1, the $R$-matrix elements must satisfy $R_\rho^{\mu\nu} R_\rho^{\nu\mu} = 1$. Such an $R$-matrix can always be found for a gauge theory [3]. We will see in section 5 and Appendix A that, with the $R$-matrix defined above, we can properly evaluate the matrix elements of the plaqette operator acting on the local Hilbert space shown in Figure 5 in the Fourier-transformed three-dimensional GT model, just as what was done in the WW model [5].

The data consisting a label set $L_G$, quantum dimension $\tilde{d} : L_G \to \mathbb{C}$ (where $\tilde{d}_\mu = \beta_\mu d_\mu$), $6j$-symbol $F : L_G^6 \to \mathbb{C}$, and an $R$-matrix form a unitary braided fusion category (UBFC) Rep($G$), which is the input data of the Fourier-transformed three-dimensional GT model with input data $G$. Therefore, as the UBFC Rep($G$) is given, we can define the Fourier transform of the total Hilbert space through (39) and the local Hilbert space Fourier transform (37). Note that although the $6j$-symbol $F$ of the category Rep($G$) can be uniquely determined through the convention introduced in subsection 3.1, multiple solutions may still exist for the $R$-matrix. From a single GT model with input data $G$, we can then construct a series of Fourier transformations with different $R$ matrix, resulting in a series of Fourier-transformed model, corresponding to the different choices of the braiding structure for the representation category Rep($G$). Nevertheless, these Fourier-transfomed models are all physically equivalent to the original model, and hence are also equivalent to each other. (For a more detailed argument, see the end of Section 5.2.) Therefore, we will not specify the braiding structure of Rep($G$). We verify that (39) preserves the orthonormality and completeness in Appendix B. Some examples for the data $\{L_G, \tilde{d}, F, R\}$ corresponding to the finite group $G$ are given in Appendix C.

## 3.3 Fourier transform of the boundary vertex operators

We are now ready to study how a boundary vertex operator $\overline{A_\nu^{\mathrm{GT}}}$ acts on a local basis state $|\Psi_{s m_s}\rangle$. That is, we need to find the Fourier-transformed version $\widetilde{A_\nu^{\mathrm{GT}}}$ of $\overline{A_\nu^{\mathrm{GT}}}$. Since the group-basis and the rewritten rep-basis $\{|\Psi_{s m_s}\rangle\}$ are both orthonormal and complete in the local Hilbert space,

we can compute the action of $\overline{\tilde{A}_v^{\mathrm{GT}}}$ by inserting resolution of identity:

$$
\begin{aligned}
\overline{\tilde{A}_v^{\mathrm{GT}}}|\Psi_{sm_s}\rangle &= \sum_{\substack{\mu'\nu'\rho'\eta'\lambda'\\ \alpha'\beta'\gamma's'm_{s'}\\ n_{\mu'}n_{\rho'}n_{\eta'}m_{\nu'}m_{\lambda'}}} |\Psi'_{s'm_{s'}}\rangle\langle\Psi'_{s'm_{s'}}|\sum_{jikgh\in G}\overline{A_v^{\mathrm{QD}}}|jikgh\rangle\langle jikgh|\Psi^{sm_s}\rangle \\
&= \sum_{\substack{\mu'\nu'\rho'\eta'\lambda'\\ \alpha'\beta'\gamma's'm_{s'}\\ n_{\mu'}n_{\rho'}n_{\eta'}m_{\nu'}m_{\lambda'}}} |\Psi'_{s'm_{s'}}\rangle\langle\Psi'_{s'm_{s'}}|\sum_{jikgh\in G}\frac{1}{|K|}\sum_{x\in K}|xj,i\bar{x},xk,xg,h\bar{x}\rangle\langle jikgh|\Psi^{sm_s}\rangle \\
&= \sum_{\substack{\mu'\nu'\rho'\eta'\lambda'\\ \alpha'\beta'\gamma's'm_{s'}\\ n_{\mu'}n_{\rho'}n_{\eta'}m_{\nu'}m_{\lambda'}}} \sum_{jikgh\in G}\frac{1}{|K|}\sum_{x\in K}\left(\langle\Psi'_{s'm_{s'}}|xj,i\bar{x},xk,xg,h\bar{x}\rangle\langle jikgh|\Psi_{sm_s}\rangle\right)|\Psi'_{s'm_{s'}}\rangle .
\end{aligned}
$$

We leave the simplification of the two inner products in the above expression in Appendix B but present the result here:

$$
\overline{\tilde{A}_v^{\mathrm{GT}}}|\Psi_{sm_s}\rangle = \sum_{m'_s}\frac{1}{|K|}\sum_{x\in K}D_{m_sm'_s}^s(x)|\Psi_{sm'_s}\rangle \equiv \sum_{m'_s}(P_K^s)_{m_sm'_s}|\Psi_{sm'_s}\rangle . \tag{42}
$$

We observe that the Fourier-transformed vertex operator $\overline{\tilde{A}_v^{\mathrm{GT}}}$ is automatically diagonalized in the entire local Hilbert space of the considered vertex spanned by $|\Psi_{sm_s}\rangle$ but the small subspace — the representation space $V_s$ of $s$, which is spanned by $m_s$. Here, $P_K^s = (1/|K|)\sum_{x\in K}D^s(x)$ is a projector in $V_s$. As the eigenvalues of a projector must be 0 or 1, a linear transformation $V_s \to V_s, |m_s\rangle \mapsto |\tilde{m}_s\rangle$ can be applied to diagonalize the matrix $P_K^s$. This transformation will also transform the basis $|\Psi_{sm_s}\rangle$ to $|\Psi_{s\tilde{m}_s}\rangle$. In such a basis, (42) can be simplified as

$$
\overline{\tilde{A}_v^{\mathrm{GT}}}|\Psi_{s\tilde{m}_s}\rangle = P_K^s|\Psi_{s\tilde{m}_s}\rangle = \delta_{(s,\tilde{m}_s)\in L_A}|\Psi_{s\tilde{m}_s}\rangle , \tag{43}
$$

where the set $L_A$ collects all the +1 eigenstates of $\overline{\tilde{A}_v^{\mathrm{GT}}}$ or its representation $P_K^s$. More precisely, we can write

$$
L_A := \{(s,\alpha_s)|P_K^s|\Psi_{s\alpha_s}\rangle = |\Psi_{s\alpha_s}\rangle, s\in L_G\} . \tag{44}
$$

The states $|\Psi_{s\tilde{m}_s}\rangle$ with $(s,\tilde{m}_s)\notin L_A$ are zero eigenstates of $\overline{\tilde{A}_v^{\mathrm{GT}}}$, and are thus excited states. These excitations emerging at the end of the dangling edges, labeled by a pair $(s,\tilde{m}_s)$, are point-like charge excitations at the boundary.

In bulk, the Fourier transform of the vertex operator is similar to the derivation above when $K = G$. The bulk vertex operator projects all degrees of freedom on the tail to the trivial representation, that is, $A_v|\Psi_{sm_s}\rangle = \delta_{s=0}|\Psi_{sm_s}\rangle$. Therefore, point-like charge excitations in the bulk are on those non-trivial tails.

## 3.4 Fourier transform of the boundary edge operators

We then investigate how the Fourier-transformed boundary edge operator $\overline{\tilde{C}_e^{\mathrm{GT}}}$ acts on a local basis state in the rewritten rep-basis of the Fourier-transformed GT model with gapped boundaries. According to (7), $\overline{C_e^{\mathrm{GT}}}$ acts on the edge between two vertices. Consequently, in the rep-basis, a local basis state on which $\overline{\tilde{C}_e^{\mathrm{GT}}}$ acts would involve the two end vertices of edge $e$. In what follows, we denote the local basis states in the group-basis acted by a boundary edge operator as:

$$
|\Psi_l\rangle := \left| \vcenter{\hbox{$\begin{smallmatrix} i & h \\ g & j & l & y \\ & & x & z & w \end{smallmatrix}$}} \right\rangle , \tag{45}
$$

and denote the local basis states in the rewritten rep-basis as:

$$|\Psi_{s\tilde{m}_s,r\tilde{m}_r}^{\nu\pi\phi\lambda}\rangle = \left| \quad \right\rangle , \tag{46}$$

where we have omitted some Latin indices, and the indices $\tilde{m}_r$ and $\tilde{m}_s$ diagonalize the boundary vertex operators acting on the two vertices in (46). In the rewritten rep-basis, the boundary edge operator is not diagonalized only on the subspace spanned by the degrees of freedom $\tilde{m}_s, s, \nu, \pi, \phi, \lambda, r$, and $\tilde{m}_r$. Thus, the boundary edge operator can be viewed as an operator acting on the open plaquette outlined by these edges.

Similar to the Fourier transform of vertex operators, we can compute the matrix elements of $\overline{\tilde{C}_e^{\text{GT}}}$ in the local basis by inserting resolution of identity:

$$
\begin{aligned}
\langle\Psi_{s'\tilde{m}_s',r'\tilde{m}_r'}^{\nu'\pi'\phi'\lambda'}|\overline{\tilde{C}_e^{\text{GT}}}|\Psi_{s\tilde{m}_s,r\tilde{m}_r}^{\nu\pi\phi\lambda}\rangle &= \langle\Psi_{s'\tilde{m}_s',r'\tilde{m}_r'}^{\nu'\pi'\phi'\lambda'}| \sum_{ghijlxyzw\in G} \overline{C_e^{\text{GT}}}|\Psi_l\rangle\langle\Psi_l|\Psi_{s\tilde{m}_s,r\tilde{m}_r}^{\nu\pi\phi\lambda}\rangle \\
&= \sum_{ghijlxyzw\in G} \delta_{l\in K}\langle\Psi_{s'\tilde{m}_s',r'\tilde{m}_r'}^{\nu'\pi'\phi'\lambda'}|\Psi_l\rangle\langle\Psi_l|\Psi_{s\tilde{m}_s,r\tilde{m}_r}^{\nu\pi\phi\lambda}\rangle \\
&= \sum_{ghijlxyzw\in G} \sum_{(t,\alpha_t)\in L_A} \frac{|K|}{|G|}d_t D_{\alpha_t\alpha_t}^t(l)\langle\Psi_{s'\tilde{m}_s',r'\tilde{m}_r'}^{\nu'\pi'\phi'\lambda'}|\Psi_l\rangle\langle\Psi_l|\Psi_{s\tilde{m}_s,r\tilde{m}_r}^{\nu\pi\phi\lambda}\rangle ,
\end{aligned}
\tag{47}
$$

where use is made of that

$$\delta_{l\in K} = \sum_{(t,\alpha_t)\in L_A} \frac{|K|}{|G|}d_t D_{\alpha_t\alpha_t}^t(l). \tag{48}$$

The proof of the the above equation and the simplification of the two inner products in (47) are left in Appendix B. Here, we just write down the result:

$$
\begin{aligned}
&\langle\Psi_{s'\tilde{m}_s',r'\tilde{m}_r'}^{\nu'\pi'\phi'\lambda'}|\overline{\tilde{C}_e^{\text{GT}}}|\Psi_{s\tilde{m}_s,r\tilde{m}_r}^{\nu\pi\phi\lambda}\rangle \\
&= \sum_{(t,\alpha_t)\in L_A} \frac{|K|}{|G|}\tilde{d}_t\tilde{v}_\nu\tilde{v}_\pi\tilde{v}_\phi\tilde{v}_\lambda\tilde{v}_s\tilde{v}_r[\tilde{v}']R_\phi^{\pi\sigma}\overline{R_{\phi'}^{\pi'\sigma}}C_{s'\tilde{m}_{s'}}^{st;\tilde{m}_s\alpha_t}C_{tr';\alpha_t\tilde{m}_{r'}}^{r\tilde{m}_r} \\
&\qquad\qquad \times G_{ts'^*\nu'}^{\mu^*\nu s^*}G_{t\nu'^*\pi'}^{\rho\pi\nu^*}G_{t\pi'^*\phi'}^{\sigma^*\phi\pi^*}G_{t\phi'^*\lambda'}^{\eta\lambda\phi^*}G_{\delta r^*r'}^{t\lambda'^*\lambda} \\
&= \sum_{(t,\alpha_t)\in L_A} \frac{|K|}{|G|}\tilde{d}_t\tilde{v}_\nu\tilde{v}_\pi\tilde{v}_\phi\tilde{v}_\lambda\tilde{v}_s\tilde{v}_r[\tilde{v}']R_\phi^{\pi\sigma}\overline{R_{\phi'}^{\pi'\sigma}}(C_{r'^*t^*r,\tilde{m}_{r'^*}\tilde{m}_{t^*}\tilde{m}_r})^*(\Omega^{r'^*})_{\tilde{m}_{r'}\tilde{m}_{r'^*}}^{-1}(\Omega^{t^*})_{\alpha_t\tilde{m}_{t^*}}^{-1} \\
&\qquad\qquad \times (C_{s'^*st,\tilde{m}_{s'^*}\tilde{m}_s\alpha_t})^*(\Omega^{s'^*})_{\tilde{m}_s'\tilde{m}_{s'^*}}^{-1}G_{ts'^*\nu'}^{\mu^*\nu s^*}G_{t\nu'^*\pi'}^{\rho\pi\nu^*}G_{t\pi'^*\phi'}^{\sigma^*\phi\pi^*}G_{t\phi'^*\lambda'}^{\eta\lambda\phi^*}G_{\delta r^*r'}^{t\lambda'^*\lambda} ,
\end{aligned}
\tag{49}
$$

where $\tilde{v}_\mu = \sqrt{\tilde{d}_\mu}$, and $\tilde{v}_\mu\cdots\tilde{v}_\nu[\tilde{v}'] := \tilde{v}_\mu\cdots\tilde{v}_\nu\tilde{v}_{\mu'}\cdots\tilde{v}_{\nu'}$ is introduced as a shorthand notation. The local state $|\Psi_{s\tilde{m}_s,r\tilde{m}_r}^{\nu\pi\phi\lambda}\rangle$ may be a $+1$ or zero eigenstate of the boundary vertex operators acting on the relevant vertices. In general, we would like to study the matrix elements of $\overline{\tilde{C}_e^{\text{GT}}}$ in the local basis states free of any charge excitations. This is accomplished by acting the boundary vertex operators on the states $|\Psi_{s\tilde{m}_s,r\tilde{m}_r}^{\nu\pi\phi\lambda}\rangle$ to project out all the charge excitations. Equivalently, we can simply replace the indices $\tilde{m}_s$ and $\tilde{m}_r$ in (49) by $\alpha_s$ and $\alpha_r$ in (44) and

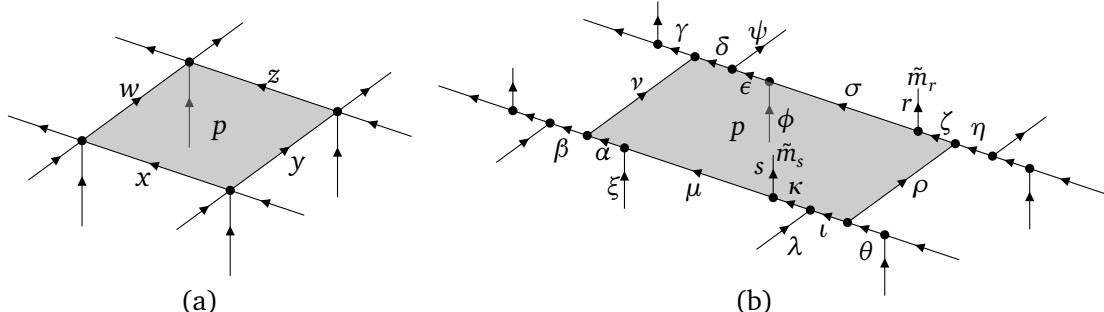

$$(a) \qquad\qquad\qquad (b)$$

Figure 6: The local Hilbert space where the boundary plaquette operators $\overline{B}_p^{\text{GT}}$ and $\overline{\tilde{B}}_p^{\text{GT}}$ act on. (a) is the group basis, denoted as $|wxyz\rangle$, and (b) is the rewritten rep-basis, denoted as $|\Psi_{r\tilde{m}_r,s\tilde{m}_s}^{\mu\alpha\nu\delta\epsilon\sigma\zeta\rho\iota\kappa}\rangle$. Here we have ignored all the group element labels and irreducible representation labels which will not appear in the matrix elements of the boundary plaquette operator.

obtain

$$
\langle\Psi_{s'\alpha_{s'},r'\alpha_{r'}}^{\nu'\pi'\phi'\lambda'}|\overline{\tilde{C}_e^{\text{GT}}}|\Psi_{s\alpha_s,r\alpha_r}^{\nu\pi\phi\lambda}\rangle
$$

$$
= \sum_{(t,\alpha_t)\in L_A} \frac{|K|}{|G|} \tilde{d}_t \tilde{v}_\nu \tilde{v}_\pi \tilde{v}_\phi \tilde{v}_\lambda \tilde{v}_s \tilde{v}_r [\tilde{v}'] R_\phi^{\pi\sigma} \overline{R_{\phi'}^{\pi'\sigma}} (C_{r'^*t^*r,\tilde{m}_{r'^*}\tilde{m}_{t^*}\alpha_r})^* (\Omega^{r'^*})_{\alpha_{r'}\tilde{m}_{r'^*}}^{-1} (\Omega^{t^*})_{\alpha_t\tilde{m}_{t^*}}^{-1}
$$

$$
\times (C_{s'^*st,\tilde{m}_{s'^*}\alpha_s\alpha_t})^* (\Omega^{s'^*})_{\alpha'_s\tilde{m}_{s'^*}}^{-1} G_{ts'^*\nu'}^{\mu^*\nu s^*} G_{t\nu'^*\pi'}^{\rho\pi\nu^*} G_{t\pi'^*\phi'}^{\sigma^*\phi\pi^*} G_{t\phi'^*\lambda'}^{\eta\lambda\phi^*} G_{\delta r^*r'}^{t\lambda'^*\lambda} \tag{50}
$$

$$
= \sum_{(t,\alpha_t)\in L_A} \frac{|K|}{|G|} \tilde{v}_t (\tilde{v}_\nu \tilde{v}_\pi \tilde{v}_\phi \tilde{v}_\lambda [\tilde{v}']) u_s u_r [u'] R_\phi^{\pi\sigma} \overline{R_{\phi'}^{\pi'\sigma}} G_{ts'^*\nu'}^{\mu^*\nu s^*} G_{t\nu'^*\pi'}^{\rho\pi\nu^*} G_{t\pi'^*\phi'}^{\sigma^*\phi\pi^*} G_{t\phi'^*\lambda'}^{\eta\lambda\phi^*} G_{\delta r^*r'}^{t\lambda'^*\lambda}
$$

$$
\times f_{r'^*\alpha_{r'}t^*\alpha_t r\alpha_r} f_{s'^*\alpha_{s'}s\alpha_s t\alpha_t},
$$

where in the last equality we define a map $f : L_A \times L_A \times L_A \to \mathbb{C}$ as

$$
f_{c^*\alpha_c a\alpha_a b\alpha_b} = \sum_{\tilde{m}_{c^*}} u_a u_b u_c (C_{c^*ab;\tilde{m}_{c^*}\alpha_a\alpha_b})^* (\Omega^{c^*})_{\alpha_c\tilde{m}_{c^*}}^{-1}, \quad \text{with} \quad (a,\alpha_a),(b,\alpha_b),(c,\alpha_c)\in L_A, \tag{51}
$$

where $u_a = \sqrt{\tilde{v}_a}$.

## 3.5 Fourier transform of the boundary plaquette operators

Finally, we discuss the Fourier transform of the boundary plaquette operators $\overline{\tilde{B}}_p^{\text{GT}}$. The local Hilbert space where $\overline{\tilde{B}}_p^{\text{GT}}$ acts on is shown in Figure 6.

We can calculate the group elements of $\overline{\tilde{B}}_p^{\text{GT}}$ through a similar procedure of the calculation in the previous section, resulting in

$$
\langle\Psi_{r'\tilde{m}_r',s'\tilde{m}_s'}^{\mu'\alpha'\nu'\delta'\epsilon'\sigma'\zeta'\rho'\iota'\kappa'}|\overline{\tilde{B}}_p^{\text{GT}}|\Psi_{r\tilde{m}_r,s\tilde{m}_s}^{\mu\alpha\nu\delta\epsilon\sigma\zeta\rho\iota\kappa}\rangle
$$

$$
= \sum \delta_{xw\bar{z}\bar{y},e} \langle\Psi_{r'\tilde{m}_r',s'\tilde{m}_s'}^{\mu'\alpha'\nu'\delta'\epsilon'\sigma'\zeta'\rho'\iota'\kappa'}|xyzw\rangle\langle xyzw|\Psi_{r\tilde{m}_r,s\tilde{m}_s}^{\mu\alpha\nu\delta\epsilon\sigma\zeta\rho\iota\kappa}\rangle \tag{52}
$$

$$
= \sum_{t\in L_G,m_t} \frac{1}{|G|} d_t D_{m_t m_t}^t (xw\bar{z}\bar{y}) \langle\Psi_{r'\tilde{m}_r',s'\tilde{m}_s'}^{\mu'\alpha'\nu'\delta'\epsilon'\sigma'\zeta'\rho'\iota'\kappa'}|xyzw\rangle\langle xyzw|\Psi_{r\tilde{m}_r,s\tilde{m}_s}^{\mu\alpha\nu\delta\epsilon\sigma\zeta\rho\iota\kappa}\rangle,
$$

where the first summation is over all the relevant group element labels in the group basis state $|xyzw\rangle$, including those that we haven't explicitly specified in Figure 6. Here, we shall focus

on the boundary degrees of freedom $(r, \tilde{m}_r)$ and $(s, \tilde{m}_s)$, which will be projected into the set $L_A$ by boundary vertex operators. We want to find out whether the Fourier-transformed boundary plaquette operators would introduce new structures on the set $L_A$, like what we have found in the Fourier transformation of the boundary edge operator where a map $f : L_A \times L_A \times L_A \to \mathbb{R}$ emerges. The answer is however negative.

Since evaluating (52) is very tedious, and since a similar procedure will be shown in section 5, we simply present the result of (52):

$$\langle \Psi' | \overline{\tilde{B}_p^{\text{GT}}} | \Psi \rangle = \sum_{t \in L_G} \tilde{d}_t \, \tilde{v}_\mu \tilde{v}_\alpha \cdots \tilde{v}_\kappa [\tilde{v}'] R_\kappa^{\mu s} \overline{R_\sigma^{\phi \epsilon}} \overline{R_\kappa^{\mu' s}} R_{\sigma'}^{\phi \epsilon'} G_{t\kappa' \iota'}^{\lambda \iota \kappa} G_{t\iota' \rho'^*}^{\theta \rho^* \iota} \cdots G_{t\mu' \kappa'}^{s \kappa \mu}, \tag{53}$$

where $|\Psi\rangle = |\Psi_{r\tilde{m}_r, s\tilde{m}_s}^{\mu \alpha \nu \delta \epsilon \sigma \zeta \rho \iota \kappa}\rangle$, $\langle \Psi'| = \langle \Psi_{r'\tilde{m}_r', s'\tilde{m}_s'}^{\mu' \alpha' \nu' \delta' \epsilon' \sigma' \zeta' \rho' \iota' \kappa'}|$, and the ellipsis between those $6j$-symbols represent other $6j$-symbols corresponding to the vertices along the boundary of plaquette $p$.

While factors $R_\kappa^{\mu s}$ and $\overline{R_{\kappa'}^{\mu' s}}$ in (53) depend on $s$, which labels the tail, these $R$-matrices do not introduce any new structures on $L_A$ because $s$ should be viewed as an element of $L_G$ here. In fact, each Fourier-transformed boundary plaquette operators $\overline{\tilde{B}_p^{\text{GT}}}$ is an identity operator within the subspace spanned by $(s, \tilde{m}_s)$ and $(r, \tilde{m}_r)$. Hence, unlike $\overline{\tilde{A}_\nu^{\text{GT}}}$ and $\overline{\tilde{C}_e^{\text{GT}}}$, the boundary plaquette operators $\overline{\tilde{B}_p^{\text{GT}}}$ do not project out any degrees of freedom on the tails attached to the relevant boundary vertices, and thus do not provide any new gapped boundary condition for the Fourier-transformed three-dimensional GT model.

So far we have Fourier-transformed and rewritten the three-dimensional GT model with gapped boundaries on a trivalent lattice $\tilde{\Gamma}$. In the sequel sections, we shall study the physics revealed by the Fourier transform.

# 4 Emergence of Frobenius algebras and boundary charge condensation

In this section, we first examine the gapped boundary condition of the Fourier-transformed GT model, then characterize the gapped boundaries by charge condensation, and finally explain why the Fourier transform makes the physics at the gapped boundaries of the GT model more explicit than it was before the transformation.

## 4.1 The gapped boundary condition of the Fourier-transformed model

In the previous section, we have found that each Fourier-transformed boundary vertex operator $\overline{\tilde{A}_\nu^{\text{GT}}}$ projects the degrees of freedom $(s, \tilde{m}_s)$ on the tail attached to the vertex $\nu$ into a set $L_A$. Along with the Fourier-transformed boundary edge operators, an emergent map $f : L_A \times L_A \times L_A \to \mathbb{R}$ appears. Nevertheless, within the subspace spanned by $(s, \tilde{m}_s)$, the relevant boundary plaquette operators behave as identity operators. Hence, the gapped boundary condition of the Fourier-transformed three-dimensional GT model can be specified by a pair $(L_A, f)$, which is determined by the input data $G$ and the boundary condition $K$ of the original GT model.

As proved in the two-dimensional case [1], the gapped boundary condition $(L_A, f)$ indeed forms a Frobenius algebra $A$, which is an object in the UBFC Rep$(G)$. Generally, an element of $L_A$ is denoted as a pair $(s, \alpha_s)$ (or simply $s\alpha_s$), where $s$ is a simple object of a UBFC $\tilde{\mathfrak{F}}$. The multiplicity of $s$ in $A$ is denoted by $|s|$ and refers to the number of different pairs $(s, \alpha_s)$ with the same $s$. The multiplication is a map $f : L_A \times L_A \times L_A \to \mathbb{C}$ that satisfies the following

associativity and non-degeneracy:

$$\sum_{c\alpha_c} f_{a\alpha_a b\alpha_b c^*\alpha_c} f_{c\alpha_c r\alpha_r s^*\alpha_s} G^{abc^*}_{rs^*t} \tilde{v}_c \tilde{v}_t = \sum_{\alpha_t} f_{a\alpha_a t\alpha_t s^*\alpha_s} f_{b\alpha_b r\alpha_r t^*\alpha_t},$$

$$f_{b\alpha_b b'^*\alpha_{b'} 0} = \delta_{bb'} \delta_{\alpha_b \alpha'_b} \beta_b,$$

(54)

where 0 is the unit element of $A$ and has multiplicity 1, i.e. $0 \equiv (0,1)$, and $\tilde{v}_c = \sqrt{\tilde{d}_c}$ with $\tilde{d}_c$ the quantum dimension of element $c$. That the Frobenius algebra $A$ defined above is an object of the corresponding UBFC $\mathfrak{F}$ is understood by writing $A$ as $A = \bigoplus_{s|s\alpha_s \in L_A} s^{\oplus |s|}$, which is generally a non-simple object in $\mathfrak{F}$. For computational convenience, one may also write $A = \bigoplus_{(s,\alpha_s) \in L_A} s_{\alpha_s}$, explicitly treating different appearances of $s$ as distinct elements of $A$.

Therefore, we conclude that the gapped boundaries of the Fourier-transformed three-dimensional GT model are specified by the Frobenius algebra $A_{G,K} = (L_A, f)_{G,K}$, which is determined by the input data $G$ and the boundary condition $K$ of the original GT model through (44) and (51).

## 4.2 Charge condensation at the gapped boundary

We will subsequently see that the boundary input data $A_{G,K}$ of the Fourier-transformed model precisely mirrors the charge condensation at the boundary. Recalling (43), each pair $(s, \alpha_s)$ labels a local $+1$ eigenstate $|\Psi_{s\alpha_s}\rangle$ at vertex $v$ of the projector $\tilde{A}^{\mathrm{GT}}_v$. All eigenstates with matching hidden labels form a subspace within the $d_s$-dimensional representation space $V_s$. The dimension $|s|$ of this subspace is determined by the count of $(s, \alpha_s)$ pairs with the same $s$. Following the emergence of the Frobenius algebra, this dimension $|s|$ is identified as the multiplicity of $s$ found in $A_{G,K}$. This identification has a strong connection with the charge condensation in three-dimensional topological orders. We quickly summarize this relationship as follows.

In two-dimensional topological orders, there exist only point-like excitations called anyons, including charges, fluxes, and dyons. The anyon condensation in two-dimensional topological orders has been extensively studied recently [17–22, 24–28, 47, 48]. On the other hand, point-like excitations in three-dimensional topological orders are pure charges, with additional loop-like and string-like excitations present. Nevertheless, the investigation of charge condensation in three-dimensional topological orders remains limited to specific cases [34].

Generally, in an $n$-dimensional topological order $\mathcal{C}$, certain types of elementary excitations may condense and cause a phase transition that takes the topological order to a simpler child topological order $\mathcal{U}$. In an extreme case, $\mathcal{U}$ could be merely a vacuum (a symmetry-protected topological phase precisely speaking) [28, 47, 49], rendering the original topological order entirely broken. This process can also be viewed from the perspective of creating a gapped domain wall that separates $\mathcal{C}$ and $\mathcal{U}$ [26]. When $\mathcal{U}$ is a vacuum, we say that certain types of elementary excitations of $\mathcal{C}$ can move to and condense at the gapped boundary between $\mathcal{C}$ and the vacuum.

Although a general classification of loop-like excitations in three-dimensional topological orders is very complicated [50–52], except for those in three-dimensional GT models with finite Abelian gauge groups, the charge excitations in the three-dimensional GT model are always classified by irreducible representation of the input data $G$. We can thus investigate the charge condensation in three-dimensional topological orders through the three-dimensional GT model with gapped boundaries with bulk input data $G$ and boundary input data $K \subseteq G$. Let us consider the situation where $K = \{e\}$, known as the rough boundary condition. Recalling Eqs.(43) and (44), all irreducible representations in $L_G$ must also be included in $L_A$. This means that $L_A = \{(s, \alpha_s) | s \in L_G, \alpha_s = 1, 2, \ldots, d_s\}$. Since the charge excitations in the bulk are labeled by $s \in L_G$, and since each pair $(s, \alpha_s)$ is an independent component of $A_{G,K}$, we say

that the pure charge $s$ divides into $d_s$ parts, each of which condenses at the boundary. Consequently, the multiplicity of the charge $s$ in the boundary condensate is $d_s = |s|$, which coincides with the multiplicity of $s$ in the Frobenius algebra $A_{G,K}$. In situations where $K$ is a nontrivial subgroup, it could be that for some $s$, only a subset $\{(s, \alpha_s) | \alpha = 1, 2, \dots, |s| < d_s\} \subset L_A$ is found. That is, even though the pure charge $s$ splits into $d_s$ pieces, only $|s|$ pieces contribute to the boundary condensate. For example, letting $G = S_3$ and $K = \mathbb{Z}_2$, we have $L_A = \{(0, 1), (2, 1)\}$, where 0 denotes the trivial representation and 2 denotes the two-dimensional irreducible representation of $S_3$. We thus have $|2| = 1 < d_2 = 2$, which means that the charge labeled by 2 splits into two pieces, only one of which labeled by $(2, 1)$ condenses at the boundary.

From the above discussion, we can see that in three-dimensional topological orders, the charge condensation at the boundary is completely described by the boundary condition. This statement can also be understood through the layer construction. The layer construction offers another interpretation for condensation of charge excitations at the boundary of a three-dimensional topological order [32, 33, 53]. Here, three-dimensional topological orders are achieved by sequentially stacking two-dimensional topological orders. For instance, a three-dimensional GT model with input data $G$ can be built by stacking the QD model with input data $G$ as well. The layers are then glued together by condensing specific types of quasiparticle pairs between them. Nevertheless, at the final layer of the two-dimensional topological order, which is the boundary of the three-dimensional topological order, excitations at the boundary are allowed to condense separately. Different boundary conditions correspond to different condensates at the boundary.

## 5 Mapping the three-dimensional GT model to WW model

The two-dimensional QD model with group $G$ as input data has already been proven to be identified with an LW model with UFC Rep$(G)$ as input data, via the Fourier transform [1]. A natural question arises: Will the Fourier transform map the GT model, i.e., the three-dimensional version of the QD model, to the WW model, i.e., the three-dimensional version of the LW model? The answer is yes. In this section, we will show that the Fourier transform defined in subsection 3.2 indeed maps the three-dimensional GT model with input data $G$ to the WW model with UBFC Rep$(G)$ as input data. Since the original WW model does not have a boundary Hamiltonian, and since there has not been any fully systematic construction of the gapped boundary of the WW model, we shall consider the bulk first and then the boundary. We also suppose that all representations of $G$ are self-dual for simplicity, as was done in [5]. It is straightforward to generalize to the case where non-self-dual representations exist. Details of the definition of the WW model can be found in Appendix A.

### 5.1 Mapping the bulk Hilbert space

By (33) and the basis rewriting in Figure 4, one can always rewrite a Fourier-transformed six-valent vertex as a trivalent lattice, as shown in Figure 8, with an extra tail attached to the vertex. The edges are labeled by irreducible representations $\mu \in L_G$ of group $G$. As the representations of a finite group $G$ form a UBFC Rep$(G)$, the sub-Hilbert space of the Fourier-transformed three-dimensional GT model where all degrees of freedom on the tails are restricted to the trivial representation, denoted as $\tilde{\mathcal{H}}_0^{GT}$, is the same as the Hilbert space of the WW model without charge excitations. Since (33) requires that each vertex in the rewritten rep-basis of the Fourier-transformed three-dimensional GT model satisfies the fusion rules, all the states in $\tilde{\mathcal{H}}_0^{GT}$ are already the $+1$ eigenstates of the vertex operators in the WW model. Charge excitations can be studied in the larger Hilbert space that contains non-trivial tails.

Nevertheless, in the following we focus on the plaquette operator and work in the Hilbert space $\tilde{\mathcal{H}}_0^{\mathrm{GT}}$.

## 5.2 Mapping the bulk Hamiltonian

In order to Fourier-transform the bulk plaquette operator of the GT model, we consider two plaquette states, $|\Psi_{abcdpqruvw}\rangle$ defined in (A.4) and state $|g_1 g_2 g_3 g_4\rangle$ defined by

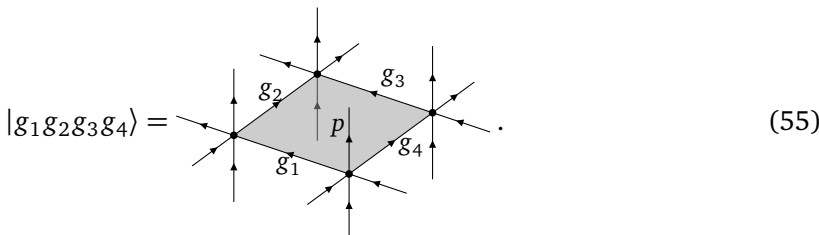

$$|g_1 g_2 g_3 g_4\rangle = \quad . \tag{55}$$

As discussed earlier, state (A.4) can also be viewed as a state in the Hilbert space of the Fourier-transformed GT model. Therefore, by (37), we can construct the inner product between the two states above as

$$\langle g_1 g_2 g_3 g_4 | \Psi_{a\cdots w}\rangle = \mathcal{N} \times v_a \cdots v_w \qquad , \tag{56}$$

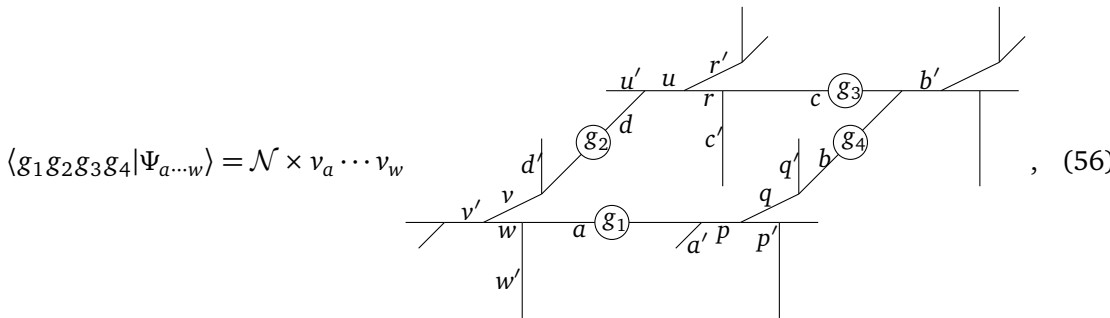

where we have collected all unimportant coefficients into $\mathcal{N}$, and we also neglect all the Latin indices $m_{q'},\ldots$ Note that

$$\mathcal{N} = \frac{v_{p'} v_{q'} v_{b'} v_{r'} v_{u'} v_{d'} v_{v'} v_{w'} v_{a'}}{|G|^2}$$

will not be affected by the action of the plaquette operator.

Inserting resolution of identity, we can write the action of the plaquette operator $\tilde{B}_p^{\mathrm{GT}}$ of the Fourier-transformed GT model as

$$\begin{aligned}
&\tilde{B}_p^{\mathrm{GT}} |\Psi_{abcdpqruvw}\rangle \\
&= \sum_{g_1 g_2 g_3 g_4 \in G} B_p^{\mathrm{GT}} |g_1 g_2 g_3 g_4\rangle \langle g_1 g_2 g_3 g_4 | \Psi_{abcdpqruvw}\rangle \\
&= \sum_{g_1 g_2 g_3 g_4 \in G} \delta_{g_1 g_2 \bar{g}_3 \bar{g}_4, e} |g_1 g_2 g_3 g_4\rangle \langle g_1 g_2 g_3 g_4 | \Psi_{abcdpqruvw}\rangle \\
&= \sum_{g_1 g_2 g_3 g_4 \in G} \sum_{s \in L_G, m_t} \frac{1}{|G|} d_s D_{m_s m_s}^s (g_1 g_2 \bar{g}_3 \bar{g}_4) |g_1 g_2 g_3 g_4\rangle \langle g_1 g_2 g_3 g_4 | \Psi_{abcdpqruvw}\rangle.
\end{aligned} \tag{57}$$

In order to compare $\tilde{B}_p^{\text{GT}}$ to $B_p^{\text{WW}}$, by means of our graphical tool, we re-express (57) as

$$\tilde{B}_p^{\text{GT}}|\Psi_{abcdpqruvw}\rangle \tag{58}$$

$$= \sum_{g_1g_2g_3g_4 \in G} \sum_{s \in L_G} \frac{\tilde{d}_s}{|G|} \mathcal{N} v_a \cdots v_w \quad \text{(graph)} \quad |g_1g_2g_3g_4\rangle,$$

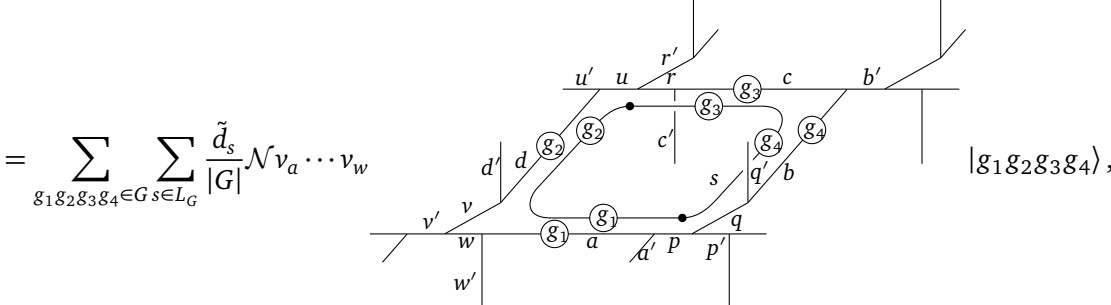

where use is made of

$$\text{(circular diagram)} = \beta_s D_{m_s m_s}^s (g_1 g_2 \bar{g}_3 \bar{g}_4). \tag{59}$$

It is the braiding structure (41) of the representation category $\text{Rep}(G)$ that enables us to put the loop $s$ everywhere we want in the entire graph containing all edges. Note that here the crossing between loop $s$ and edges $c'$ and $q'$ are inevitable if we want to place loop $s$ along this plaquette correctly.

To evaluate the graph in the equation above, we introduce the convention

$$C_{\mu,m_\mu}^{\mu 0,n_\mu} C_{0\nu,m_\nu}^{\nu,n_\nu} = \frac{1}{\tilde{v}_\mu \tilde{v}_\nu} \delta_{n_\mu m_\mu} \delta_{n_\nu m_\nu}, \tag{60}$$

which can be expressed graphically as

$$\text{(diagram: } \mu, \nu \text{ lines)} = \tilde{v}_\mu \tilde{v}_\nu \text{ (diagram: } \mu, \nu \text{ with dotted line)},$$

where the dotted line is graced by the trivial representation. It is easy to prove that convention (60) is consistent with the first equation in (24) and the $F$-move (27). We can then apply $F$-moves in the graph in (58) and obtain

$$\tilde{B}_p^{\text{GT}}|\Psi_{abcdpqruvw}\rangle = \sum_{g_1g_2g_3g_4 \in G} \sum_{s \in L_G} \frac{\tilde{d}_s}{|G|} R_q^{q'b} \overline{R_c^{c'r}} \overline{R_{q''}^{q'b''}} R_{c''}^{c'r''} F_{sa''p''}^{a'pa} F_{sp''q''}^{p'qp} \cdots F_{sw''a''}^{w'aw}$$

$$\times \tilde{d}_a \tilde{d}_b \tilde{d}_c \tilde{d}_d \mathcal{N} v_a \cdots v_w$$

$$\times \quad \text{(graph)} \quad |g_1g_2g_3g_4\rangle. \tag{61}$$

Then, applying the second equation in (24) and using the intertwiner property of $3j$-symbols, we find that the graph in the above equation is equal to

$$\frac{\langle g_1 g_2 g_3 g_4 | \Psi_{a''b''\cdots w''} \rangle}{\tilde{d}_a \tilde{d}_b \tilde{d}_c \tilde{d}_d \times \mathcal{N} v_{a''} \cdots v_{w''}} \, .$$

The results above enable us to write the matrix elements of the Fourier-transformed plaquette operators of the three-dimensional GT model explicitly as

$$
\begin{aligned}
\langle \Psi_{a''\cdots w''} | \tilde{B}_p^{\mathrm{GT}} | \Psi_{a\cdots w} \rangle &= \sum_{s \in L_G} \frac{\tilde{d}_s}{|G|} \frac{v_a \cdots v_w}{v_{a''} \cdots v_{w''}} R_q^{q'b} \overline{R_c^{c'r}} \, \overline{R_{q''}^{q'b''}} R_{c''}^{c'r''} F_{sa''p''}^{a'pa} F_{sp''q''}^{p'qp} \cdots F_{sw''a''}^{w'aw} \\
&= \sum_{s \in L_G} \frac{\tilde{d}_s}{|G|} \tilde{v}_a \cdots \tilde{v}_w [\tilde{v}''] R_q^{q'b} \overline{R_c^{c'r}} \, \overline{R_{q''}^{q'b''}} R_{c''}^{c'r''} G_{sa''p''}^{a'pa} G_{sp''q''}^{p'qp} \cdots G_{sw''a''}^{w'aw} \, ,
\end{aligned}
\tag{62}
$$

where the second equality is due to that $F_{\eta\kappa\rho}^{\mu\nu\lambda} = \tilde{d}_\rho G_{\eta\kappa\rho}^{\mu\nu\lambda}$ (28) is used.

Taking into account that $D^2 = \sum_{s \in L_G} \tilde{d}_s^2 = \sum_{s \in L_G} d_s^2 = |G|$, we find that $\tilde{B}_p^{\mathrm{GT}}$ is exactly the same as $B_p^{\mathrm{WW}}$. Therefore, as far as bulk is concerned, after the Fourier transform and basis rewriting, and finally projecting all the degrees of freedom on the tails to the trivial representation, the three-dimensional GT model with input data $G$ has the same Hilbert space and Hamiltonian term-by-term as the WW model with input data Rep($G$).

We can then discuss the relationship between the ground states and excited states of the three-dimensional GT model and the WW model. As the Hamiltonian and the Hilbert space of the three-dimensional GT model are exactly mapped to the Hamiltonian and the Hilbert space of the WW model, the ground states of the two models are also related by the Fourier-transform. Hence, the ground-state degeneracies of the two models are also the same, which indicates that the two models realize the same phases of matter. We then consider the exitation states. One type of bulk excitations in three-dimensional GT model is loop-like excitation, which occurs when $B_p^{\mathrm{GT}} = 0$ on a series of plaquettes forming a loop. As the Fourier-transformed plaquette operator $\tilde{B}_p^{\mathrm{GT}}$ is exactly identified with $B_p^{\mathrm{WW}}$, it follows that the excited states associated with loop-like excitations in the three-dimensional GT model analogously map to the excited states of the WW model, with loop-like excitations living at the same positions of the lattice. A similar argument also holds for charge excitations which occur when $A_v^{\mathrm{GT}} = 0$. Therefore, we conclude that the three-dimensional GT model with input data $G$ and the WW model with input data Rep($G$) exactly describe the same topological order. Moreover, different WW models with input data Rep($G$) equipped with different braiding structures are also equivalent. Starting with a GT model with input data $G$ (denoted by $\mathrm{GT}_G$), given the UFC Rep($G$) equipped with a set of $R$-symbols, we can define a Fourier transform, which maps the Hamiltonian of the GT model to the Hamiltonian of a WW model term by term. This transformation (denoted by $\mathrm{FT}_R$) is a unitary linear transformation that does not affect the spectrum of the Hamiltonian and the ground state degeneracy. Therefore, the resulting WW model (denoted by $\mathrm{WW}_{\mathrm{Rep}(G),R}$) is physically equivalent to the original GT model $\mathrm{GT}_G$. Nevertheless, if we choose another set of $R$-symbols denoted by $R'$, we will get another WW model $\mathrm{WW}_{\mathrm{Rep}(G),R'}$ via the transformation $\mathrm{FT}_{R'}$, which is also physically equivalent to the original GT model $\mathrm{GT}_G$. Therefore, the two WW models $\mathrm{WW}_{\mathrm{Rep}(G),R}$ and $\mathrm{WW}_{\mathrm{Rep}(G),R'}$ are related through the transformation $\mathrm{FT}_{R'} \circ \mathrm{FT}_R^{-1}$, and hence must be physically equivalent.

### 5.3 Mapping the boundary

Recall that as aforementioned, the gapped boundary theory of the WW model has not been fully systematically constructed before, except for smooth boundaries and some special cases.

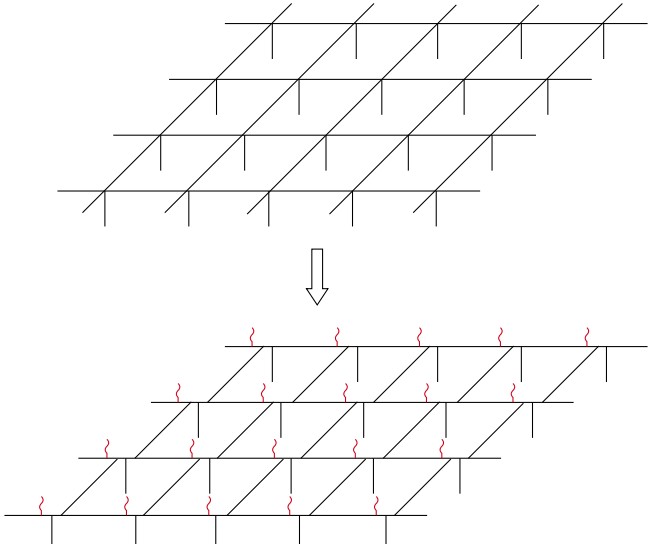

Figure 7: Upper: The original boundary lattice. Lower: The Fourier-transformed boundary lattice. Red wiggly lines are tails attached to original vertices, with degrees of freedom taking value in Frobenius algebra $A_{G,K}$. The degrees of freedom on the black straight lines still take value in $\text{Rep}(G)$.

Now that we have already mapped the bulk Hilbert space and Hamiltonian to the WW model via Fourier transform, the Fourier-transformed boundary of the three-dimensional GT model with input data $G$ also offers a full systematic construction of the gapped boundary theory of the WW model with input data $\text{Rep}(G)$. The construction is understood as follows. The boundary lattice is shown in Figure 7. The boundary Hilbert space is the tensor product of the local Hilbert spaces on those edges and tails. The black edges are labeled by the objects of the UBFC $\text{Rep}(G)$, while the tails are labeled by elements in the Frobenius algebra $A_{G,K} \in \text{Rep}(G)$. The boundary Hilbert space can thus be expressed as

$$\overline{\mathcal{H}^{\text{WW}}} = \left( \bigotimes_{e \in \partial \tilde{\Gamma}} \text{span}_{j_e \in \mathcal{R}ep(G)}\{|j_e\rangle\} \right) \otimes \left( \bigotimes_{t \in \text{boundary tails}} \text{span}_{j_t \in A_{G,K}}\{|j_t\rangle\} \right). \tag{63}$$

The boundary Hamiltonian consists of respectively the sums of boundary vertex, plaquette, and edge operators:

$$\overline{H^{\text{WW}}} = -\sum_{v \in \partial \tilde{\Gamma}} \overline{A_v^{\text{WW}}} - \sum_{p \in \partial \tilde{\Gamma}} \overline{B_p^{\text{WW}}} - \sum_{e \in \partial \Gamma} \overline{C_e^{\text{WW}}}. \tag{64}$$

The boundary vertex operator $\overline{A_v^{\text{WW}}}$ is the identity operator if the labels of the three edges (or tails) around $v$ obey the fusion rules; otherwise it is 0. The boundary plaquette operator $\overline{B_p^{\text{WW}}}$ is given by (53). The boundary edge operator $\overline{C_e^{\text{WW}}}$ acts on the local Hilbert space corresponds to edge $e$ of the original cubic lattice, as shown in (46), with its matrix elements given by (50). Since these local operators are obtained from the boundary Hamiltonian of the three-dimensional GT model via Fourier transform, they commute with each other. Therefore, the total Hamiltonian of the WW model is still exactly solvable.

## Acknowledgments

YW is grateful for the hospitality of the Perimeter Institute during his visit, where the main part of this work is done.

**Funding information** This research was supported in part by the Perimeter Institute for Theoretical Physics. Research at Perimeter Institute is supported by the Government of Canada through the Department of Innovation, Science and Economic Development and by the Province of Ontario through the Ministry of Research, Innovation and Science. YW is supported by NSFC Grant No. KRH1512711, the Shanghai Municipal Science and Technology Major Project (Grant No. 2019SHZDZX01), Science and Technology Commission of Shanghai Municipality (Grant No. 24LZ1400100), and the Innovation Program for Quantum Science and Technology (No. 2024ZD0300101). YH is supported by NSFC (Grant No. 12375001), National Key Research and Development Program of China (Grant No. 2024YFA1408900), and Zhejiang Provincial Natural Science Foundation of China (Grant No. LY23A050001).

## A  The Walker-Wang model

The Walker-Wang model is defined on a three-dimensional trivalent vertex $\Gamma$, which is deformed from a three-dimensional cubic lattice. At each six-valent vertex of the cubic lattice, the deformation is depicted in Figure 8.

The input data of the model is a UBFC, in which the string types will be labeled by Latin letters $a, b, c, \ldots \in L$, are assigned to the edges of $\Gamma$. The Hilbert space $\mathcal{H}^{\mathrm{WW}}$ of the model is spanned by all labels of the edges in the lattice $\Gamma$. The definition of UBFC also includes a set of symmetrized $6j$-symbols $G : L^6 \to \mathbb{C}$ and a set of $R$-symbols $R : L^3 \to \mathbb{C}$. The $6j$-symbols give the following basis transformation:

$$
\begin{array}{c} a \diagdown \; b \\ m \diagup \; c \\ d \end{array} \;=\; \sum_n \tilde{v}_m \tilde{v}_n G^{bam}_{dcn} \; \begin{array}{c} b \diagdown \; c \\ a \; n \\ d \end{array} , \tag{A.1}
$$

and the $R$-symbols encode the braidings:

$$
\begin{array}{c} a \diagdown \; b \\ \bigcirc \\ c \end{array} \;=\; R^{ab}_c \; \begin{array}{c} a \diagdown \; b \\ \diagup \\ c \end{array} . \tag{A.2}
$$

In the WW model, we also assume multiplicity-free in the fusion rules and self-duality of all labels, so edges in our lattices are not oriented. The Hamiltonian of the model is

$$
H = -\sum_{v \in \Gamma} A^{\mathrm{WW}}_v - \sum_{p \in \Gamma} B^{\mathrm{WW}}_p , \tag{A.3}
$$

where $v$ ranges over all vertices in the trivalent lattice $\Gamma$, and $p$ ranges over all plaquettes in $\Gamma$ which correspond to the original squares of the cubic lattice. For $|\Psi\rangle \in \mathcal{H}^{\mathrm{WW}}$, we have

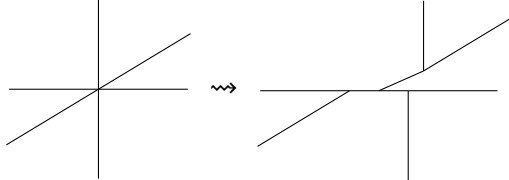

Figure 8: Deform a vertex in a three-dimensional cubic lattice to a trivalent lattice in the WW model.

(a)

(b)

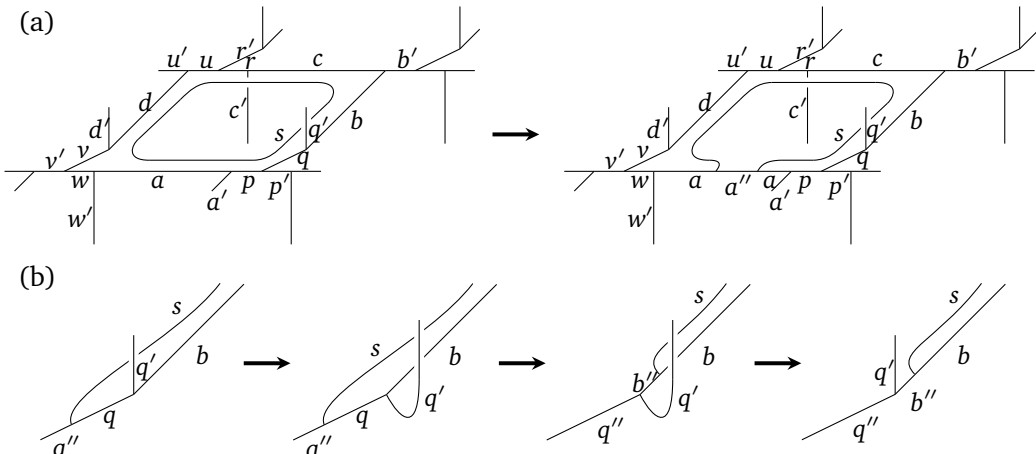

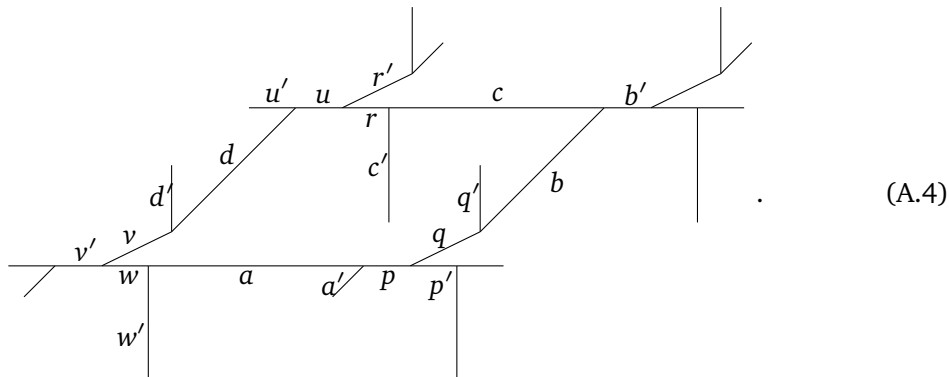

Figure 9: (a) Fusing the simple loop labeled by $s$ with the edge labeled by $a$. (b) In order to fuse the string labeled by $s$ with edge labeled by $b$, first we have to act a $R$-symbol at the vertex and then apply the basis transformation (A.1); otherwise the string $s$ will fuse with edge labeled by $q'$, which is not in the boundary of the plaquette.

$A_v|\Psi\rangle = |\Psi\rangle$ if the three labels around vertex $v$ obey the fusion rules; otherwise $A_v|\Psi\rangle = 0$. In order to derive the plaquette operator $B_p^{\text{WW}}$, we consider the following plaquette with the relevant labels:

$$ \text{(A.4)} $$

Then, analogous to the LW model, we define $B_p^{\text{WW}} = \sum_s (\tilde{d}_s/D^2) B_p^s$, where $D^2 = \sum_{s \in L} \tilde{d}_s^2$, and the action of $B_p^s$ on the state (A.4) is to fuse a simple loop labeled by $s$ with the edges around the plaquette, through the basis transformation (A.1), as shown in Figure 9(a). However, things get tricky when we encounter the vertex where $q, q'$, and $b$ meet. As shown in Figure 9(b), explicitly applying the basis transformation (A.1) results in a state where string $s$ fuses with $q'$, which is not what we want. Thus, we have to twist the vertex using (A.2) before applying (A.1), and twist the vertex back finally to recover the original lattice. The same procedure is also applied when dealing with the vertex where $c, c'$, and $r$ meet. Therefore, compared to the LW model, we get four extra $R$-symbols in the expression of the matrix elements of $B_p^s$.

Denote the state (A.4) as $|\Psi_{abcdpqruvw}\rangle$, then the explicit expression of the matrix elements of $B_p^s$ is given by

$$ \langle \Psi_{a''\cdots w''} | B_p^s | \Psi_{a\cdots w} \rangle = \tilde{v}_a \cdots \tilde{v}_w [\tilde{v}''] R_q^{q'b} \overline{R_c^{c'r}} \overline{R_{q''}^{q'b''}} R_{c''}^{c'r''} $$
$$ \times G_{sa''p''}^{a'pa} G_{sp''q''}^{p'qp} G_{sq''b''}^{q'bq} G_{sb''c''}^{b'cb} G_{sc''r''}^{c'rc} G_{sr''u''}^{r'ur} G_{su''d''}^{u'du} G_{sd''v''}^{d'vd} G_{sv''w''}^{v'wv} G_{sw''a''}^{w'aw} . \tag{A.5} $$

Note that in order to evaluate the above expression, here we use the following convention to

remove the bubble:

$$\left| \begin{array}{c} a' \\ b\ \ c \\ a \end{array} \right\rangle = \frac{\tilde{v}_b \tilde{v}_c}{\tilde{v}_a} \delta_{a'a} \left| \ a\ \right\rangle , \tag{A.6}$$

which is different from our graphical presentation (24).

## B  Some proofs

Here we prove that the rewritten rep-basis defined in (36) is orthonormal and complete.

$$\langle \Psi'_{s'm'_s} | \Psi_{sm_s} \rangle = \sum_{jikgh \in G} \langle \Psi'_{s'm'_s} | jikgh \rangle \langle jikgh | \Psi_{sm_s} \rangle \tag{B.1}$$

$$= \frac{v_\mu v_\nu v_\rho v_\eta v_\lambda v_\alpha v_\beta v_\gamma v_s [v']}{|G|^5} \sum_{jikgh \in G} \left( \begin{array}{c} \text{diagram} \end{array} \right)^* \left( \begin{array}{c} \text{diagram} \end{array} \right)$$

$$= \frac{v_\mu v_\nu v_\rho v_\eta v_\lambda v_\alpha v_\beta v_\gamma v_s [v']}{|G|^5} \sum_{jikgh \in G} \left( \begin{array}{c} \text{diagram} \end{array} \right) \left( \begin{array}{c} \text{diagram} \end{array} \right)$$

$$= \frac{v_\nu v_\lambda v_\alpha v_\beta v_\gamma v_s [v']}{|G|^2} \delta_{\mu'\mu} \delta_{\eta'\eta} \delta_{\rho'\rho} \delta_{n'_\mu n_\mu} \delta_{n'_\eta n_\eta} \delta_{n'_\rho n_\rho} \sum_{ih \in G} \left( \begin{array}{c} \text{diagram} \end{array} \right)$$

$$= \beta_\alpha v_\beta v_\gamma v_s [v'] \delta_{\mu'\mu} \delta_{\nu'\nu} \delta_{\eta'\eta} \delta_{\lambda'\lambda} \delta_{\rho'\rho} \delta_{\alpha'\alpha} \delta_{n'_\mu n_\mu} \delta_{m'_\nu m_\nu} \delta_{n'_\eta n_\eta} \delta_{m'_\lambda m_\lambda} \delta_{n'_\rho n_\rho} \sum_{m,n} \left( \begin{array}{c} \text{diagram} \end{array} \right) ,$$

where (24) is used, and the $\beta_\alpha$ comes from $v_\alpha v_{\alpha'} \delta_{\alpha'\alpha}/\tilde{d}_\alpha$. Although the ends labeled by $m$ and $n$ in the graphical presentation in the last line of the above expression are summed, they cannot be connected explicitly. In order to evaluate the above graphical presentation, we use the following trick:

$$\sum_n \cdots = \sum_n \cdots = \sum_{n'} \cdots = \sum_{n'} \cdots = \beta_\lambda \sum_{n'} \cdots = \beta_\lambda \cdots \lambda^*, \tag{B.2}$$

where the normalization condition of duality map and (25) are used. Noting that

$$\cdots = \cdots = \cdots, \tag{B.3}$$

we can simplify the graphical presentation in (B.1) as

$$\cdots = \beta_\nu \beta_\lambda \cdots \lambda^* = \frac{\beta_\nu \beta_\lambda \beta_\beta \beta_\gamma \beta_s}{d_\beta d_\gamma d_s} \delta_{\beta'\beta} \delta_{\gamma'\gamma} \delta_{s's} \delta_{m'_s m_s}. \tag{B.4}$$

Inserting the above expression into (B.1) yields

$$\langle \Psi'_{s'm'_s}|\Psi_{sm_s}\rangle = \beta_\nu \beta_\lambda \beta_\alpha \beta_\beta \beta_\gamma \beta_s \delta_{\mu'\mu} \delta_{\nu'\nu} \delta_{\eta'\eta} \delta_{\lambda'\lambda} \delta_{\rho'\rho} \delta_{\alpha'\alpha} \delta_{\beta'\beta} \delta_{\gamma'\gamma} \delta_{s's}$$
$$\times \delta_{n'_\mu n_\mu} \delta_{m'_\nu m_\nu} \delta_{n'_\eta n_\eta} \delta_{m'_\lambda m_\lambda} \delta_{n'_\rho n_\rho} \delta_{m'_s m_s}. \tag{B.5}$$

Finally, using the fact that $\beta_\mu \beta_\nu \beta_\rho = 1$ when $C_{\mu\nu\rho} \neq 0$, we have $\beta_\nu \beta_\lambda \beta_\alpha \beta_\beta \beta_\gamma \beta_s = 1$, and thus it is proved that the basis $|\Psi_{sm_s}\rangle$ is orthonormal.

The proof of completeness is as follows:

$$\sum_{\mu\cdots s} \sum_{m_\nu\cdots m_s} \langle jikgh|\Psi_{sm_s}\rangle \langle \Psi_{sm_s}|j'i'k'g'h'\rangle$$

$$= \sum_{\mu\cdots s} \sum_{m_\nu\cdots m_s} \frac{d_\mu d_\nu d_\rho d_\eta d_\lambda d_\alpha d_\beta d_\gamma d_s}{|G|^5} \left( \cdots \right) \left( \cdots \right)$$

$$= \sum_{\mu\cdots s} \sum_{m_\nu m_\lambda m_s} \frac{d_\mu d_\nu d_\rho d_\eta d_\lambda d_\alpha d_\beta d_\gamma d_s}{|G|^5}$$ (B.6)

where $\tilde{g} = g\bar{g}'$, $\tilde{k} = k\bar{k}'$, and $\tilde{j} = j\bar{j}'$. Then, note that

$$\cdots = \frac{1}{|G|}\sum_{l_1 \in G} \cdots ,$$ (B.7)

where (23) is used. Using the similar method, we can simplify (B.6) as

$$\sum_{\mu\cdots s} \sum_{m_\nu \cdots m_s} \langle jikgh|\Psi_{sm_s}\rangle\langle\Psi_{sm_s}|j'i'k'g'h'\rangle$$

$$= \sum_{\mu\cdots s} \sum_{m_\nu m_\lambda m_s} \frac{d_\mu d_\nu d_\rho d_\eta d_\lambda d_\alpha d_\beta d_\gamma d_s}{|G|^5} \sum_{l_1 l_2 l_3 l_4 \in G} \frac{1}{|G|^4}$$ (B.8)

$$\times \cdots .$$

Let us consider the first term in the graphical representation above, which reads

$$\sum_{m_\nu,k,l} D^\nu_{m_\nu k}(i) D^\nu_{kl}(l_4) D^\nu_{lm_\nu}(\bar{i}') = \operatorname{Tr} D^\nu(il_4\bar{i}') \equiv \operatorname{Tr}_\nu(\tilde{i}'l_4),$$ (B.9)

where $\tilde{i}' = \bar{i}'i$. Recalling that

$$\delta_{l,e} = \sum_{\mu \in L_G} \frac{1}{|G|} d_\mu \operatorname{Tr} D^\mu(l),$$ (B.10)

we find that the term (B.9) gives a factor of $\delta_{\tilde{i}'l_4,e}$. Similarly, the second term of (B.8) gives $\delta_{l_4,e}$ and the sixth term gives $\delta_{\tilde{h}'l_2,e}$ with $\tilde{h}' = \bar{h}'h$. The third term of (B.8) reads

$$\sum_{mnkl} D^{\gamma^*}_{mn}(l_4)(\Omega^\gamma)^{-1}_{mk} D^\gamma_{kl}(l_3)\Omega^{\gamma^*}_{nl} = \sum_{mnkl} \beta_\gamma (\Omega^{\gamma^*})^{-1}_{km} D^{\gamma^*}_{mn}(l_4)\Omega^{\gamma^*}_{nl} D^\gamma_{kl}(l_3)$$ (B.11)

$$= \sum_{kl} \beta_\gamma [D^\gamma_{kl}(l_4)]^* D^\gamma_{kl}(l_3) = \beta_\gamma \operatorname{Tr}_\gamma(\bar{l}_4 l_3),$$ (B.12)

where (10), the unitarity of $\Omega^\mu$, and the definition of $\beta_\mu$, $(\Omega^{\mu^*})^{\mathrm{T}} = \beta_\mu \Omega^\mu$ are used. Therefore, the third term gives $\beta_\gamma \delta_{\bar{l}_4 l_3,e}$ in (B.8). (B.8) thus can be further simplified as

$$\sum_{\mu \cdots s} \sum_{m_\nu \cdots m_s} \langle jikgh | \Psi_{sm_s} \rangle \langle \Psi_{sm_s} | j'i'k'g'h' \rangle$$
$$= \sum_{l_1 l_2 l_3 l_4 \in G} \beta_\alpha \beta_\beta \beta_\gamma \beta_\mu \beta_\rho \beta_\eta \times \delta_{\tilde{i}'l_4,e} \delta_{l_4,e} \delta_{\bar{l}_4 l_3,e} \delta_{\bar{l}_3 \tilde{g},e} \delta_{\bar{l}_3 l_2,e} \delta_{\bar{h}'l_2,e} \delta_{\bar{l}_2 l_1,e} \delta_{\bar{l}_1 \tilde{k},e} \delta_{\bar{l}_1 \tilde{j},e} \,. \tag{B.13}$$

Noting that $\beta_\alpha \beta_\rho \beta_\mu = 1$ and $\beta_\beta \beta_\gamma \beta_\eta = 1$, we finally get

$$\sum_{\mu \cdots s} \sum_{m_\nu \cdots m_s} \langle jikgh | \Psi_{sm_s} \rangle \langle \Psi_{sm_s} | j'i'k'g'h' \rangle = \delta_{\tilde{i}'i,e} \delta_{g\bar{g}',e} \delta_{\bar{h}'h,e} \delta_{k\bar{k}',e} \delta_{jj',e} = \langle jikgh | j'i'k'g'h' \rangle \,, \tag{B.14}$$

which implies that

$$\sum_{\mu \cdots s} \sum_{m_\nu \cdots m_s} | \Psi_{sm_s} \rangle \langle \Psi_{sm_s} | = \mathbb{1} \,, \tag{B.15}$$

i.e., the basis $| \Psi_{sm_s} \rangle$ is complete.

We then prove that the rewritten rep-basis with braided edges defined by (38) and (39) is orthonormal and complete. We define

$$| \mu \nu \rangle := \left| m_{\bar{\mu}} \underset{n_\nu}{\overset{m_\nu}{\underset{\mu}{\nu}}} n_\mu \right\rangle , \qquad | gh \rangle := \left| \underset{g}{\overset{h}{\uparrow}} \right\rangle . \tag{B.16}$$

Thus

$$\langle \mu'\nu' | \mu\nu \rangle = \sum_{g,h \in G} \langle \mu'\nu' | gh \rangle \langle gh | \mu\nu \rangle$$

$$= \sum_{g,h \in G} \frac{\nu_\mu \nu_\nu [\nu']}{|G|^2} \sum_{\rho,\rho'} \tilde{d}_\rho \tilde{d}_{\rho'} \left( \begin{array}{c} \text{diagram} \end{array} \right)^* \begin{array}{c} \text{diagram} \end{array}$$

$$= \sum_{g,h \in G} \frac{\nu_\mu \nu_\nu [\nu']}{|G|^2} \sum_{\rho,\rho'} \tilde{d}_\rho \tilde{d}_{\rho'} R^{\mu'\nu'}_{\rho'} \overline{R^{\mu\nu}_\rho} \begin{array}{c} \text{diagram} \end{array}$$

$$= \sum_{\rho,\rho'} \tilde{d}_\rho \tilde{d}_{\rho'} R^{\mu'\nu'}_{\rho'} \overline{R^{\mu\nu}_\rho} \delta_{\mu'\mu} \delta_{\nu'\nu} \delta_{m_{\mu'} m_\mu} \delta_{m_{\nu'} m_\nu} \begin{array}{c} \text{diagram} \end{array}$$

$$= \sum_{\rho,\rho'} \tilde{d}_{\rho'} \delta_{\rho'\rho} \delta_{\mu'\mu} \delta_{\nu'\nu} \delta_{m_{\mu'} m_\mu} \delta_{m_{\nu'} m_\nu} \begin{array}{c} \text{diagram} \end{array}$$

$$= \delta_{\rho'\rho} \delta_{\mu'\mu} \delta_{\nu'\nu} \delta_{m_{\mu'} m_\mu} \delta_{m_{\nu'} m_\nu} \delta_{n_{\mu'} n_\mu} \delta_{n_{\nu'} n_\nu} \,, \tag{B.17}$$

which proves the orthonormality.

To prove the completeness, we need to compute

$$\sum_{\mu,\nu}\sum_{m_\mu m_\nu n_\nu n_\mu}\langle gh|\mu\nu\rangle\langle\mu\nu|g'h'\rangle$$

$$=\sum_{\mu,\nu}\sum_{m_\mu m_\nu n_\nu n_\mu}\frac{d_\mu d_\rho}{|G|^2}\sum_{\rho,\rho'}\tilde{d}_\rho\tilde{d}_{\rho'}\ \ \left(\quad\right)^*$$

$$=\sum_{\mu,\nu}\sum_{m_\mu m_\nu n_\nu n_\mu}\frac{d_\mu d_\rho}{|G|^2}\sum_{\rho,\rho'}\tilde{d}_\rho\tilde{d}_{\rho'}\overline{R_\rho^{\mu\nu}}R_{\rho'}^{\mu\nu}$$

$$=\sum_{\mu,\nu}\sum_{m_\mu m_\nu}\frac{d_\mu d_\rho}{|G|^2}\sum_{\rho,\rho'}\tilde{d}_\rho\tilde{d}_{\rho'}\overline{R_\rho^{\mu\nu}}R_{\rho'}^{\mu\nu}$$

$$=\sum_{\mu,\nu}\sum_{m_\mu m_\nu}\frac{d_\mu d_\rho}{|G|^2}D^\mu_{m_\mu m_\mu}(g\bar{g}')D^\nu_{m_\nu m_\nu}(h\bar{h}')$$

$$=\delta_{g\bar{g}',e}\delta_{h\bar{h}',e}=\delta_{gg'}\delta_{hh'}=\langle gh|g'h'\rangle\,.$$

Thus

$$\sum_{\mu,\nu}\sum_{m_\mu m_\nu n_\mu n_\nu}|\mu\nu\rangle\langle\mu\nu|=\mathbb{1}\,,\tag{B.18}$$

i.e., the basis $|\mu\nu\rangle$ is complete.

Details of the action of the operators $\overline{\tilde{A}_\nu^{\mathrm{GT}}}$ are as follows.

$$\overline{\tilde{A}_\nu^{\mathrm{GT}}}|\Psi_{sm_s}\rangle$$

$$=\sum_{\substack{\mu'\cdots s'\\ n'_\mu\cdots m'_s}}\sum_{jikgh\in G}\frac{1}{|K|}\sum_{x\in K}\left(\langle\Psi'_{s'm_{s'}}|xj,i\bar{x},xk,xg,h\bar{x}\rangle\langle jikgh|\Psi_{sm_s}\rangle\right)|\Psi'_{s'm_{s'}}\rangle$$

$$=\sum_{\substack{\mu'\cdots s'\\ n'_\mu\cdots m'_s}}\sum_{jikgh\in G}\frac{1}{|K|}\sum_{x\in K}\frac{\nu_\mu\nu_\nu\nu_\rho\nu_\eta\nu_\lambda\nu_\alpha\nu_\beta\nu_\gamma\nu_s[\nu']}{|G|^5}$$

$$\times \left(\text{diagram}\right)^* \left(\text{diagram}\right) |\Psi'_{s'm_{s'}}\rangle$$

$$= \sum_{\substack{\mu'\cdots s' \\ n'_\mu\cdots m'_s}} \sum_{jikgh\in G} \frac{1}{|K|} \sum_{x\in K} \frac{\nu_\mu \nu_\nu \nu_\rho \nu_\eta \nu_\lambda \nu_\alpha \nu_\beta \nu_\gamma \nu_s[\nu']}{|G|^5}$$

$$\times \left(\text{diagram}\right)^* \left(\text{diagram}\right) |\Psi'_{s'm_{s'}}\rangle$$

$$= \sum_{\substack{\mu'\cdots s' \\ n'_\mu\cdots m'_s}} \sum_{jikgh\in G} \frac{1}{|K|} \sum_{x\in K} \frac{\nu_\mu \nu_\nu \nu_\rho \nu_\eta \nu_\lambda \nu_\alpha \nu_\beta \nu_\gamma \nu_s[\nu']}{|G|^5}$$

$$\times \left(\text{diagram}\right) \left(\text{diagram}\right) |\Psi'_{s'm_{s'}}\rangle .$$

In the third equality the fact that 3$j$-symbols are intertwiners is used. Applying the great orthogonality theorem and the trick introduced in (B.2), we have

$$\overline{\tilde{A}^{GT}_\nu}|\Psi_{sm_s}\rangle = \sum_{\substack{\mu'\cdots s' \\ n'_\mu\cdots m'_s}} \delta_{\mu'\mu}\cdots\delta_{\lambda'\lambda}\delta_{n'_\mu n_\mu}\cdots\delta_{m'_\nu m_\nu} \frac{1}{|K|} \sum_{x\in K} \beta_\lambda \beta_\nu \left(\text{diagram}\right) |\Psi'_{s'm'_s}\rangle$$

$$= \sum_{m'_s} \frac{1}{|K|} \sum_{x \in K} \beta_\lambda \beta_\nu \beta_\alpha \beta_\beta \beta_\gamma \beta_s D^s_{m_s m'_s}(\bar{x}) |\Psi_{sm'_s}\rangle$$

$$= \sum_{m'_s} \frac{1}{|K|} \sum_{x \in K} D^s_{m_s m'_s}(x) |\Psi_{sm'_s}\rangle . \tag{B.19}$$

Details of the action of the operators $\overline{\tilde{C}^{\text{GT}}_e}$ are as follows. Recall that

$$\langle \Psi^{\nu'\pi'\phi'\lambda'}_{s'\tilde{m}'_s, r'\tilde{m}'_r} | \overline{\tilde{C}^{\text{GT}}_e} | \Psi^{\nu\pi\phi\lambda}_{s\tilde{m}_s, r\tilde{m}_r} \rangle = \sum_{g \cdots w \in G} \sum_{(t,\alpha_t) \in L_A} \frac{|K|}{|G|} d_t D^t_{\alpha_t \alpha_t}(l) \langle \Psi^{\nu'\pi'\phi'\lambda'}_{s'\tilde{m}'_s, r'\tilde{m}'_r} | \Psi_l \rangle \langle \Psi_l | \Psi^{\nu\pi\phi\lambda}_{s\tilde{m}_a, r\tilde{m}_r} \rangle . \tag{B.20}$$

Similar to the calculation in section 5, we can firstly evaluate $D^t_{\alpha_t \alpha_t}(l) \langle \Psi_l | \Psi^{\nu\pi\phi\lambda}_{s\tilde{m}_a, r\tilde{m}_r} \rangle$, which reads

$$D^t_{\alpha_t \alpha_t}(l) \langle \Psi_l | \Psi^{\nu\pi\phi\lambda}_{s\tilde{m}_s, r\tilde{m}_r} \rangle = \frac{\nu_\nu \nu_\pi \nu_\phi \nu_\lambda \nu_s \nu_r \nu_\mu \nu_\rho \nu_\sigma \nu_\eta \nu_\delta}{\sqrt{|G|}} \quad , \tag{B.21}$$

where some edges and Latin indices irrelevant to the calculation are omitted. Note that the braiding between the lines $t$ and $\sigma$ is unavoidable if we want to evaluate the expression correctly. Then we can use the convention (60) and apply $F$-moves to the graph in the equation above and obtain

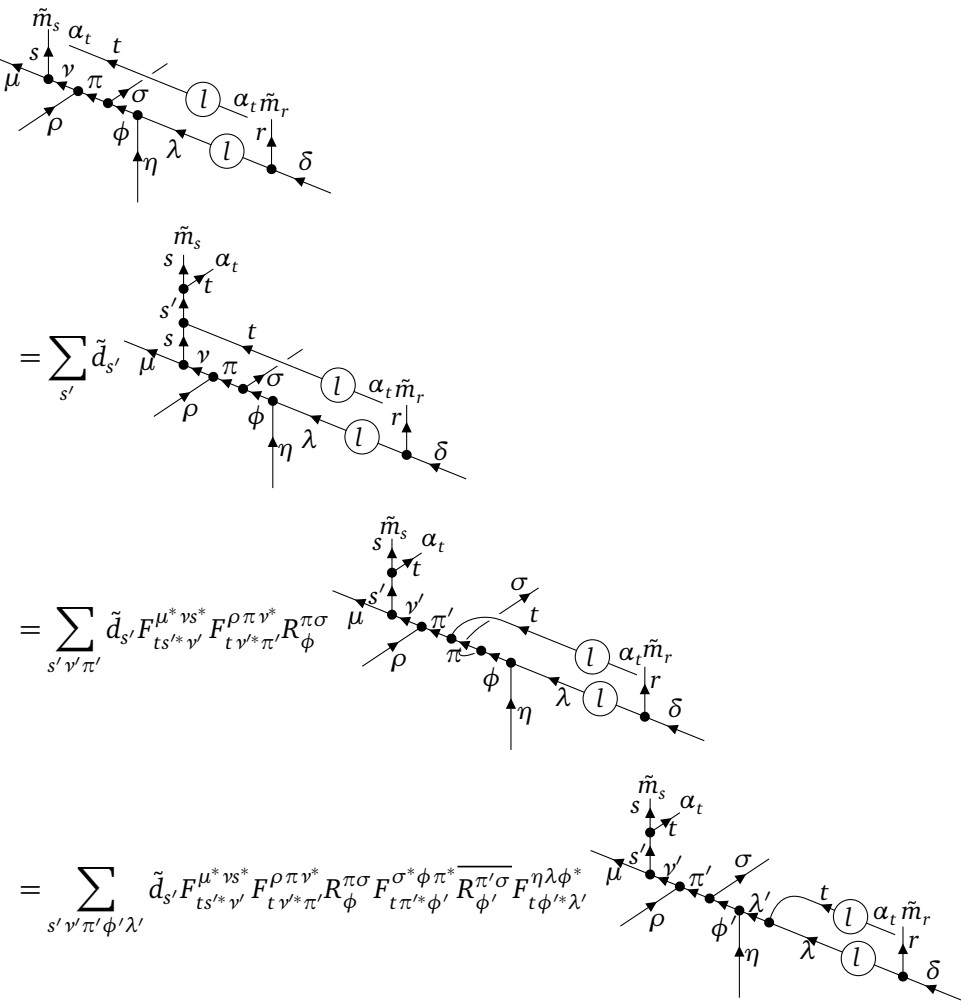

$$= \sum_{s'\nu'\pi'\phi'\lambda'} \tilde{d}_{s'} F^{\mu^*\nu s^*}_{ts'^*\nu'} F^{\rho\pi\nu^*}_{t\nu'^*\pi'} R^{\pi\sigma}_{\phi} F^{\sigma^*\phi\pi^*}_{t\pi'^*\phi'} \overline{R^{\pi'\sigma}_{\phi'}} F^{\eta\lambda\phi^*}_{t\phi'^*\lambda'} \tilde{d}_{\lambda'}$$

$$\times$$

$$= \sum_{s'\nu'\pi'\phi'\lambda'r'} \tilde{d}_{s'} F^{\mu^*\nu s^*}_{ts'^*\nu'} F^{\rho\pi\nu^*}_{t\nu'^*\pi'} R^{\pi\sigma}_{\phi} F^{\sigma^*\phi\pi^*}_{t\pi'^*\phi'} \overline{R^{\pi'\sigma}_{\phi'}} F^{\eta\lambda\phi^*}_{t\phi'^*\lambda'} F^{t\lambda'^*\lambda}_{\delta r^*r'}$$

$$= \sum_{s'\nu'\pi'\phi'\lambda'r'} \tilde{d}_{s'} F^{\mu^*\nu s^*}_{ts'^*\nu'} F^{\rho\pi\nu^*}_{t\nu'^*\pi'} R^{\pi\sigma}_{\phi} F^{\sigma^*\phi\pi^*}_{t\pi'^*\phi'} \overline{R^{\pi'\sigma}_{\phi'}} F^{\eta\lambda\phi^*}_{t\phi'^*\lambda'} F^{t\lambda'^*\lambda}_{\delta r^*r'} C^{st;\tilde{m}_s\alpha_t}_{s'\tilde{m}_{s'}} C^{t^*r;\tilde{m}_{t^*}\tilde{m}_r}_{r'\tilde{m}_{r'}} (\Omega^t)^{-1}_{\tilde{m}_{t^*}\alpha_t}$$

$$\times \frac{\sqrt{|G|}}{\nu_{\nu'}\nu_{\pi'}\nu_{\phi'}\nu_{\lambda'}\nu_{s'}\nu_{r'}\nu_{\mu}\nu_{\rho}\nu_{\sigma}\nu_{\eta}\nu_{\delta}} \langle \Psi_l | \Psi^{\nu'\pi'\phi'\lambda'}_{s'\tilde{m}_{s'},r'\tilde{m}_{r'}} \rangle. \tag{B.22}$$

Substituting the result above back to (B.20) and using the completeness and orthonormality of our basis, we get

$$\langle \Psi^{\nu'\pi'\phi'\lambda'}_{s'\tilde{m}'_s,r'\tilde{m}'_r} | \overline{\tilde{C}^{\mathrm{GT}}_e} | \Psi^{\nu\pi\phi\lambda}_{s\tilde{m}_s,r\tilde{m}_r} \rangle$$

$$= \sum_{(t,\alpha_t)\in L_A} \frac{|K|}{|G|} d_t \nu_\nu \nu_\pi \nu_\phi \nu_\lambda \nu_s \nu_r [\nu']^{-1} \tilde{d}_{s'} \tilde{d}_{\nu'} \tilde{d}_{\pi'} \tilde{d}_{\phi'} \tilde{d}_{\lambda'} \tilde{d}_{r'}$$

$$\times R^{\pi\sigma}_{\phi} \overline{R^{\pi'\sigma}_{\phi'}} G^{\mu^*\nu s^*}_{ts'^*\nu'} G^{\rho\pi\nu^*}_{t\nu'^*\pi'} G^{\sigma^*\phi\pi^*}_{t\pi'^*\phi'} G^{\eta\lambda\phi^*}_{t\phi'^*\lambda'} G^{t\lambda'^*\lambda}_{\delta r^*r'}$$

$$\times (C_{sts'^*;\tilde{m}_s\alpha_t\tilde{m}_{s'^*}})^* (\Omega^{s'})^{-1}_{\tilde{m}_{s'^*}\tilde{m}_{s'}} (C_{t^*rr'^*;\tilde{m}_{t^*}\tilde{m}_r\tilde{m}_{r'^*}})^* (\Omega^t)^{-1}_{\tilde{m}_{t^*}\alpha_t} (\Omega^{r'})^{-1}_{\tilde{m}_{r'^*}\tilde{m}_{r'}}$$

$$= \sum_{(t,\alpha_t)\in L_A} \frac{|K|}{|G|} \tilde{d}_t \nu_\nu \nu_\pi \nu_\phi \nu_\lambda \nu_s \nu_r [\nu']^{-1} \tilde{d}_{s'} \tilde{d}_{\nu'} \tilde{d}_{\pi'} \tilde{d}_{\phi'} \tilde{d}_{\lambda'} \tilde{d}_{r'}$$

$$\times R^{\pi\sigma}_{\phi} \overline{R^{\pi'\sigma}_{\phi'}} G^{\mu^*\nu s^*}_{ts'^*\nu'} G^{\rho\pi\nu^*}_{t\nu'^*\pi'} G^{\sigma^*\phi\pi^*}_{t\pi'^*\phi'} G^{\eta\lambda\phi^*}_{t\phi'^*\lambda'} G^{t\lambda'^*\lambda}_{\delta r^*r'}$$

$$\times (C_{s'^*st;\tilde{m}_{s'^*}\tilde{m}_s\alpha_t})^* (\Omega^{s'^*})^{-1}_{\tilde{m}_{s'}\tilde{m}_{s'^*}} (C_{r'^*t^*r;\tilde{m}_{r'^*}\tilde{m}_{t^*}\tilde{m}_r})^* (\Omega^{t^*})^{-1}_{\alpha_t\tilde{m}_{t^*}} (\Omega^{r'^*})^{-1}_{\tilde{m}_{r'}\tilde{m}_{r'^*}}. \tag{B.23}$$

Finally, let us check:

$$\nu_\nu \nu_\pi \nu_\phi \nu_\lambda \nu_s \nu_r [\nu']^{-1} \tilde{d}_{s'} \tilde{d}_{\nu'} \tilde{d}_{\pi'} \tilde{d}_{\phi'} \tilde{d}_{\lambda'} \tilde{d}_{r'} = \nu_\nu \nu_\pi \nu_\phi \nu_\lambda \nu_s \nu_r [\nu'] \beta_{\nu'} \beta_{\pi'} \beta_{\phi'} \beta_{\lambda'} \beta_{r'} \beta_{s'}. \tag{B.24}$$

Keeping mind mind that

$$\beta_\mu = \beta_s \beta_\nu = \beta_{s'} \beta_{\nu'} = \sqrt{\beta_s \beta_\nu \beta_{s'} \beta_{\nu'}},$$
$$\beta_\delta = \beta_\lambda \beta_r = \beta_{\lambda'} \beta_{r'} = \sqrt{\beta_\lambda \beta_r \beta_{\lambda'} \beta_{r'}}, \tag{B.25}$$
$$\beta_\sigma = \beta_\pi \beta_\phi = \beta_{\pi'} \beta_{\phi'} = \sqrt{\beta_\pi \beta_\phi \beta_{\pi'} \beta_{\phi'}},$$

we have

$$\nu_\nu \nu_\pi \nu_\phi \nu_\lambda \nu_s \nu_r [\nu'] \beta_{\nu'} \beta_{\pi'} \beta_{\phi'} \beta_{\lambda'} \beta_{r'} \beta_{s'} = \tilde{\nu}_\nu \tilde{\nu}_\pi \tilde{\nu}_\phi \tilde{\nu}_\lambda \tilde{\nu}_s \tilde{\nu}_r [\tilde{\nu}'], \tag{B.26}$$

which completes our proof of (49).

The proof of (48) is as follows: multiplying the both sides of (48) by $D^t_{\tilde{m}_t \tilde{m}'_t}(\bar{l})$ and then summing over $l \in G$ will give

$$\frac{1}{|K|} \sum_{l \in G} \delta_{l \in K} D^t_{\tilde{m}_t \tilde{m}'_t}(\bar{l}) = \sum_{(t, \alpha_t) \in L_A} \frac{d_t}{|G|} \sum_{l \in G} D^t_{\alpha_t \alpha_t}(l) D^t_{\tilde{m}_t \tilde{m}'_t}(\bar{l})$$

$$= \sum_{(t, \alpha_t) \in L_A} \delta_{\alpha_t \tilde{m}_t} \delta_{\alpha_t \tilde{m}'} \equiv \delta_{(t, \tilde{m}_t) \in L_A} \delta_{\tilde{m}_t \tilde{m}'_t} .$$

Recalling the definition of $L_A$, we find that the LHS of the equation above is

$$\frac{1}{|K|} \sum_{l \in G} \delta_{l \in K} D^t_{\tilde{m}_t \tilde{m}'_t}(\bar{l}) = \frac{1}{|K|} \sum_{l \in K} D^t_{\tilde{m}_t \tilde{m}'_t}(l) = \delta_{(t, \tilde{m}_t) \in L_A} \delta_{\tilde{m}_t \tilde{m}'_t} = \text{RHS} .$$

Thus (48) is proved. Requiring $K = \{e\}$ in (48) gives (B.10).

## C  Examples

### C.1  $G$ is Abelian

When $G$ is Abelian, $L_G$ has a group structure and is isomorphic to $G$ itself. The fusion rule is given by

$$\mu \otimes \nu = \mu + \nu , \tag{C.1}$$

where $+$ is the group multiplication of $G$. For all $\mu \in L_G$, we have $\mu^* = -\mu$ and $\tilde{d}_\mu = 1$. The $3j$-symbols and $6j$-symbols are given by

$$C_{\mu\nu\rho} = \delta_{\mu\nu\rho} , \quad \text{and} \quad G^{\mu\nu\lambda}_{\eta\kappa\rho} = \delta_{\mu\nu\lambda} \delta_{\eta\kappa\lambda^*} \delta_{\nu\eta\rho^*} \delta_{\mu\rho\kappa} , \tag{C.2}$$

where $\delta_{\mu\nu\rho} = 1$ if $\mu + \nu + \rho = 0$ (Here 0 denotes the identity element of the group $G$.) and $\delta_{ijk} = 0$ otherwise. Finally, $R$-symbols can always be trivial, i.e.,

$$R^{\mu\nu}_\rho = 1 , \tag{C.3}$$

for all $\mu + \nu + \rho = 0$.

To study the boundary condition, we then set $G = \mathbb{Z}_{n_1} \times \mathbb{Z}_{n_2} \times \cdots$. The group elements of $G$ can be written as $g = (g_1, g_2, \ldots)$ where $g_i = 0, 1, \ldots, n_i - 1$, and the irreducible representations of $G$ can be written as $\mu = (\mu_1, \mu_2, \ldots)$ where $\mu_i = 0, 1, \ldots, n_i - 1$. The representation matrix can be explicitly expressed as

$$D^\mu(g) = \exp\left[ 2\pi i \left( \frac{\mu_1 g_1}{n_1} + \frac{\mu_2 g_2}{n_2} + \ldots \right) \right] . \tag{C.4}$$

Then, for subgroup $K$, the boundary vertex operator is given by

$$\delta_{s \in L_A} = \frac{1}{|K|} \sum_{k \in K} \exp\left[ 2\pi i \left( \frac{s_1 k_1}{n_1} + \frac{s_2 k_2}{n_2} + \ldots \right) \right] . \tag{C.5}$$

Moreover, if $G = \mathbb{Z}_n$ and $K = \mathbb{Z}_m$, such that $m | n$, we have

$$\delta_{s \in L_A} = \frac{1}{m} \sum_{k=0}^{m-1} \exp\left( 2\pi i \frac{ks}{m} \right) = \frac{1 - \exp(2\pi i s)}{m[1 - \exp(2\pi i s/m)]} , \tag{C.6}$$

where the element $k \in \mathbb{Z}_m$ is labeled as $kn/m$ in $\mathbb{Z}_n$. Therefore, only when $s = jm$, $j \in \mathbb{N}$, we have $\delta_{s \in L_A} = 1$. Thus we have $L_A = 0, m, 2m, \ldots, n - m$ and

$$A_{\mathbb{Z}_n, \mathbb{Z}_m} = (L_A, \delta_{abc}) \cong \mathbb{Z}_{n/m} = \mathbb{Z}_m/\mathbb{Z}_n. \tag{C.7}$$

In general, not all Frobenius algebras in $\text{Rep}(\mathbb{Z}_{n_1} \times \mathbb{Z}_{n_2} \times \cdots)$ are given by (C.5). Those Frobenius algebras beyond (C.5) correspond to gapped boundaries with non-trivial twist.

## C.2   $G = D_3$

The dihedral group $D_3$ has six group elements $\{1, s, r, r^2, sr, sr^2\}$, where $s^2 = 1$ and $r^3 = 1$. The irreducible representations of $D_3$ are the trivial representation, the sign representation, and the two-dimensional definition representation, denoted by $\mathbf{0}, \mathbf{1}, \mathbf{2}$, respectively. More explicitly, we have

$$
\begin{aligned}
D^{\mathbf{0}}(g) &= 1, & \forall g \in D_3, \\
D^{\mathbf{1}}(e) &= D^{\mathbf{1}}(r) = 1, & D^{\mathbf{1}}(s) &= -1, \\
D^{\mathbf{2}}(e) &= \begin{pmatrix} 1 & 0 \\ 0 & 1 \end{pmatrix}, & D^{\mathbf{2}}(r) &= \frac{1}{2}\begin{pmatrix} -1 & -\sqrt{3} \\ \sqrt{3} & -1 \end{pmatrix}, & D^{\mathbf{2}}(s) &= \frac{1}{2}\begin{pmatrix} -1 & 0 \\ 0 & 1 \end{pmatrix}.
\end{aligned}
\tag{C.8}
$$

These representations are obviously real and self-dual (which means that those duality maps defined by (10) are just identity matrices), and hence $\beta_\mu = 1$ for all $\mu \in L_{D_3}$. The quantum dimensions of these irreducible representations are given by $\tilde{d}_{\mathbf{0}} = \tilde{d}_{\mathbf{1}} = 1$, $\tilde{d}_{\mathbf{2}} = 2$. The non-trivial fusion rules of $\text{Rep}(D_3)$ are

$$\mathbf{1} \otimes \mathbf{1} = \mathbf{0}, \qquad \mathbf{1} \otimes \mathbf{2} = \mathbf{2}, \qquad \mathbf{2} \otimes \mathbf{2} = \mathbf{0} \oplus \mathbf{1} \oplus \mathbf{2} \oplus \mathbf{2}, \tag{C.9}$$

and hence $\delta_{\mathbf{000}} = \delta_{\mathbf{011}} = \delta_{\mathbf{022}} = \delta_{\mathbf{122}} = \delta_{\mathbf{222}} = 1$. The non-vanishing 3j-symbols of $D_3$ are

$$
\begin{aligned}
C_{\mathbf{000}} &= C_{\mathbf{011}} = 1, \\
C_{\mathbf{0}2_m 2_n} &= \frac{1}{\sqrt{2}}\begin{pmatrix} 1 & 0 \\ 0 & 1 \end{pmatrix}_{mn}, & C_{\mathbf{1}2_m 2_n} &= \frac{1}{\sqrt{2}}\begin{pmatrix} 0 & i \\ -i & 0 \end{pmatrix}_{mn}, \\
C_{\mathbf{2}_1 2_m 2_n} &= \frac{1}{2}\begin{pmatrix} 0 & 1 \\ 1 & 0 \end{pmatrix}_{mn}, & C_{\mathbf{2}_2 2_m 2_n} &= \frac{1}{2}\begin{pmatrix} 1 & 0 \\ 0 & -1 \end{pmatrix}_{mn}.
\end{aligned}
\tag{C.10}
$$

Symmetrized 6j-symbols of $\text{Rep}(D_3)$ can then be calculated from (29). The nonzero 6j-symbols are

$$
\begin{aligned}
G_{\mathbf{000}}^{\mathbf{000}} &= G_{\mathbf{111}}^{\mathbf{000}} = G_{\mathbf{011}}^{\mathbf{000}} = 1, \\
G_{\mathbf{222}}^{\mathbf{000}} &= G_{\mathbf{222}}^{\mathbf{011}} = \frac{1}{\sqrt{2}}, \\
G_{\mathbf{022}}^{\mathbf{022}} &= G_{\mathbf{122}}^{\mathbf{022}} = G_{\mathbf{222}}^{\mathbf{022}} = G_{\mathbf{122}}^{\mathbf{122}} = -G_{\mathbf{222}}^{\mathbf{122}} = \frac{1}{2}.
\end{aligned}
\tag{C.11}
$$

All other non-zero 6j-symbols are obtained through the tetrahedral symmetry. Given the data $\{\tilde{d}, \delta, G\}$, the R-matrix $R : L_{D_3}^3 \to \mathbb{C}$ can be solved from the hexagon equations:

$$
\begin{aligned}
\sum_\mu \tilde{d}_\mu G_{\gamma\epsilon\phi}^{\alpha\beta\mu^*} G_{\beta\epsilon^*\mu}^{\gamma\alpha\delta^*} R_{\mu\gamma}^\epsilon &= G_{\beta\epsilon^*\phi}^{\alpha\gamma\delta^*} R_{\alpha\gamma}^\delta R_{\beta\gamma}^\phi, \\
\sum_\mu \tilde{d}_\mu G_{\gamma\alpha\mu}^{\epsilon^*\beta\delta} G_{\beta\gamma\phi}^{\epsilon^*\alpha\mu} R_{\alpha\delta}^\epsilon &= G_{\alpha\gamma\phi}^{\epsilon^*\beta\delta} R_{\alpha\gamma}^\delta R_{\alpha\beta}^\phi,
\end{aligned}
\tag{C.12}
$$

where all Greek indices belong to $L_{D_3}$. The non-trivial $R$-matrix is

$$R^{\mathbf{1}}_{\mathbf{22}} = -1 \,. \tag{C.13}$$

The data $\{\tilde{d}, \delta, G, R\}$ describes the unitary braided fusion category $\mathrm{Rep}(D_3)$.

Using (51), we can obtain four emergent Frobenius algebras corresponding to the four distinct subgroups of $D_3$:

- $K = D_3$, $L_A = \{\mathbf{0}\}$, $A = \mathbf{0}$, $f_{\mathbf{000}} = 1$.

- $K = \mathbb{Z}_3$, $L_A = \{\mathbf{0}, \mathbf{1}\}$, $A = \mathbf{0} \oplus \mathbf{1}$, $f_{\mathbf{011}} = 1$.

- $K = \mathbb{Z}_2$, $L_A = \{\mathbf{0}, \mathbf{2}_1\}$, $A = \mathbf{0} \oplus \mathbf{2}$, $f_{\mathbf{2}_1\mathbf{2}_1\mathbf{2}_1} = 2^{-1/4}$, $f_{\mathbf{02}_1\mathbf{2}_1} = 1$.

- $K = \{e\}$, $L_A = \{\mathbf{0}, \mathbf{1}, \mathbf{2}_1, \mathbf{2}_2\}$, $A = \mathbf{0} \oplus \mathbf{1} \oplus \mathbf{2} \oplus \mathbf{2}$. Non-trivial multiplications are given by

$$f_{\mathbf{12}_1\mathbf{2}_2} = -\mathrm{i}, \qquad f_{\mathbf{12}_2\mathbf{2}_1} = \mathrm{i}, \qquad f_{\mathbf{2}_2\mathbf{2}_1\mathbf{2}_1} = 2^{-1/4}, \qquad f_{\mathbf{2}_2\mathbf{2}_2\mathbf{2}_2} = -2^{-1/4} \,.$$

  Note that $f_{\mathbf{2}_1\mathbf{2}_1\mathbf{2}_1} = 0$ because $C_{\mathbf{2}_1\mathbf{2}_1\mathbf{2}_1} = 0$.

More explicitly, let us consider the case $K = \mathbb{Z}_2 = \{e, s\}$ which has been briefly discussed in Section 4.2. We have

$$P^{\mathbf{2}}_{\mathbb{Z}_2} = \frac{1}{2} \sum_{x=e,s} D^{\mathbf{2}}(x) = \frac{1}{2} \begin{pmatrix} 0 & 0 \\ 0 & 1 \end{pmatrix} \,. \tag{C.14}$$

Hence, only half of the two-dimensional space $V_{\mathbf{2}}$ or half of the charge-$\mathbf{2}$ condenses at the boundary.

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
