# Peer review of "Fourier-transformed gauge theory models of three-dimensional topological orders with gapped boundaries"

_SciPost Physics, doi:SciPost Phys. 19, 018 (2025)_

## Round 1 · Referee Report · Anonymous (Referee 1) · 2024-8-26

Strengths

  1. The paper is well-organized and clearly explains the correspondence between lattice quantum field theory and the state-sum construction of TQFT.
  2. The Frobenius algebra-based boundary theory is extended to the 3d model.
  3. An explicit Fourier transformation between the G-lattice gauge theory and the Walker-Wang model is provided.

Weaknesses

There are many typos that need to be revised.

Report

In this work, the author investigates the gapped boundary theory of a 3d lattice model of topological order. Similar to the 2d case [JHEP01(2018)134], the construction is based on the Frobenius algebra within the input unitary fusion category $\mathrm{Rep}(G)$. Through the Fourier transform, which reformulates the original model into the fusion basis, they argue that the 3d lattice gauge theory is equivalent to the Walker-Wang model. This generalizes the well-established result in 2d, where lattice gauge theory is equivalent to the Levin-Wen string-net model.

The results are interesting and the paper is well-written. I believe this work meets the publication criteria for Sci-Post.

Requested changes

Before publication, I believe the paper requires thorough proofreading, as there are some typos that need to be addressed, and there may potentially be more.

P 5, Line -2 of the first paragraph, "oriented cubic lattice $\Gamma$ which boundaries", "which" should be "with".

P 6, Line 2 below Eq.(7), "operators are all commute with each other".

P 7, Caption of Fig 2, "that terminat on the boundary".

P 7, Line above Eq. (11), "the duality map and its inverse has presentations".

P 10, Line -4, "Additionaly" should be "Additionally".

P 13, Caption of Fig 4, "by contracting there indices", "with an free end".

P 15, Line 2 of the second paragraph, "larger Hilber space".

P 15, Caption of Fig 5, "as are edges labeleds by".

P 27, Above Fig 9, "comparing with the LW model."

Recommendation

Ask for minor revision

---

## Round 1 · Referee Report · Anonymous (Referee 2) · 2024-9-10

Strengths

  1. The paper presents all of the technical details of the calculations
  2. The calculations appear to be technically correct
  3. The results are potentially useful for advancing knowledge in the field

Weaknesses

  1. The authors need to give a clearer explanation of what can be learned from the correspondence that they establish
  2. There are a number of points (see below) that are poorly explained or confusing.
  3. The paper presents no concrete examples. These could be useful first to clarify certain points in an accessible way (see 2), but also to illustrate more concretely what can be learned from this approach that has not been described elsewhere in the literature. For example, perhaps the authors can use their approach to identify a class of gapped boundaries for non-abelian gauge models that were not previously known.
  4. The paper has no discussion or conclusion.

Report

The authors use the Fourier transform of general groups to re-write a conventional 3D gauge theory model with gauge group G in a basis where edges are instead labeled by group representations. They discuss how to do this both in the bulk and at the boundary, as well as how to make an explicit connection between the model in the representation basis and a 3D topological lattice model due to Walker and Wang. They use this to describe in general terms some gapped boundaries of these models that have not been described previously to the best of my knowledge.

The results that this paper obtains are certainly not surprising, and its main contributions to the literature are highly technical in nature, at the level of how in detail to carry out a mapping from one basis to the other. That said, this is a case where there is merit to writing out the technical details, and I could be persuaded that this work makes a sufficiently substantive contribution to the literature on 3D topological lattice models to warrant publication in Scipost. However, in its current form I find the paper to be very heavy on technical details and light on discussion of their physical interpretation and importance. I think that the authors could do a better job of:
(1) articulating what can be learned from this formulation, other than showing that two previously known models can be mapped one onto the other
(2) clarifying what their general framework says about simple examples, such as simple abelian theories, and then working out some specific examples of e.g. interesting gapped boundaries in non-abelian models.
(3) Making the connection to known models, and especially the Walker-Wang model, clearer (I have a number of specific comments about this below.).

Requested changes

ntroduction:
— I’m confused about the comparison to lattice gauge theory. Specifically: the topological models contain a different Hilbert space, in which the Gauss’ law constraint is satisfied only energetically (i.e. in the ground state). Thus the models discussed are very similar to gauge theory models with non-dynamical matter (in which charges can appear as static excitations). However, modulo this difference, the topological models are essentially a particular limit of lattice gauge theories, in which the fluctuations of the electric field are set to 0. (This is discussed, for example, in the PhD thesis of Mark De Wilde Propitious). I am similarly confused about what the authors mean when they say that lattice gauge theory has only local excitations: because of the constraint, charge excitations can live only at the ends of Wilson lines, and the constraint that the net magnetic flux passing through a closed surface must be an integer multiple of 2 pi is present in both theories. I suggest that the authors clarify this discussion.

Section 3:
— It would be helpful if the authors commented on the connections between their Fourier transform and a change of basis in a quantum gauge theory from the group basis to a basis of eigenstates of the electric field. Intuitively, I expected them to be related, but the need to introduce a braiding suggests otherwise.
— Similarly, the authors should clarify the following. In the original model, violations of the vertex term are clearly given by representations of G, while violations of the plaquette term are given by group elements. I don’t see a discussion of the excitations in the Fourier transformed model (when tails are included). This is implicit in the construction but some readers may benefit from a more detailed discussion.
— I am not sure I understand Eq. 39. In Walker and Wang’s formulation, the R factors appear in two ways. First, in the Hamiltonian, from “twisting” an edge relative to another edge with which it shares a trivalent vertex. Second, they dictate relative phases between different edge configurations, in which the configuration effectively has a twisted loop. The picture in Eq. (58) is the one relevant to evaluating the plaquette operator. Fig. 5 appears to be illustrating a situation with a crossing of the first type, in which the R matrix should be telling us about a relative phase between two configurations in a ground state. Could the authors clarify the connection between this and Eq. 38? It is not obvious to me.
— I am also confused about how we choose R. The authors seem to be implying in the discussion below (41) that there is *a* choice of R. However, for general G, I expected there to be multiple choices. First, naively I expected that taking R to be trivial (always +1) is always allowed, and that this is the choice that would have the same spectrum as the original G theory (i.e. bulk point-like charges and line-like vortex loops). Second, I would also have expected that there are other choices of R that satisfy the hexagon equations, and that these correspond to physically different theories in which the number of point particles in the bulk is reduced. For example, when the Hilbert space on each edge is just Rep(Z_2 ) = Vec(Z_2), there are two choices for R, and these yield the 3D Toric code (with point particles and vortices) and the “3D semion model” which does not have point-like bulk excitations. The authors should address this point more thoroughly in Section 3, so that readers can understand how much freedom there is in choosing R, and how the choice of R affects the final theory. In my view this is a very central question, since it determines whether we are discussing “the” Fourier transform of the model in question, or simply “a” Fourier transform…
— Please discuss the role of R at the boundary in more detail. Again, comparing the 3D Toric code to 3D semion model, we see that the former admits two distinct gapped boundaries (which are usually thought of as flux condensing and charge condensing), while the latter apparently only admits one. Hence I would have thought that there are some stipulations on what boundary condition(s) can be chosen, and that these depend on R.

Section 4:
— In the third paragraph of Section 4, the authors should specify whether they are discussing 2+1 D or 3+1 D topological orders. They should also clarify the relationship between this paragraph and the discussion of coupled layer constructions in the following paragraph.
— Loop like excitations in 3D abelian topological orders are, however, well understood. The authors should note this and include Ref. https://journals.aps.org/prb/abstract/10.1103/PhysRevB.91.165119 .
— The authors should add a discussion of the physics of these gapped boundaries. For example, given a subset of charges that can condense, what happens to loops that are brought to the boundary?

Section 5:
— I don’t understand what the authors mean when they say: “no loss of information is
“ the WW model does not contain charge and dyon excitations, just like the LW model.” Dyons in 3D theories would require point-like monopoles carrying charge — but the discrete gauge theories discussed here don’t have magnetic monopoles. So why are we talking about dyons at all?

Recommendation

Ask for major revision

  • validity: good
  • significance: ok
  • originality: ok
  • clarity: low
  • formatting: good
  • grammar: reasonable

Author:  Siyuan Wang  on 2025-01-03  [id 5081]

(in reply to Report 2 on 2024-09-10)
Category:
answer to question

We would like to express our sincere gratitude to the editor and the referee for their insightful comments and suggestions on our manuscript. We appreciate the time and effort taken by the referees to review our manuscript and provide valuable feedback. Below are our responses to the referee's questions.

The referee writes:

Introduction:
— I'm confused about the comparison to lattice gauge theory. Specifically: the topological models contain a different Hilbert space, in which the Gauss' law constraint is satisfied only energetically (i.e. in the ground state). Thus the models discussed are very similar to gauge theory models with non-dynamical matter (in which charges can appear as static excitations). However, modulo this difference, the topological models are essentially a particular limit of lattice gauge theories, in which the fluctuations of the electric field are set to 0. (This is discussed, for example, in the PhD thesis of Mark De Wilde Propitious). I am similarly confused about what the authors mean when they say that lattice gauge theory has only local excitations: because of the constraint, charge excitations can live only at the ends of Wilson lines, and the constraint that the net magnetic flux passing through a closed surface must be an integer multiple of 2 pi is present in both theories. I suggest that the authors clarify this discussion.

Our response:

We acknowledge that this discussion is unnecessary and misleading. Therefore, we have removed this paragraph.

The referee writes:

Section 3:
— It would be helpful if the authors commented on the connections between their Fourier transform and a change of basis in a quantum gauge theory from the group basis to a basis of eigenstates of the electric field. Intuitively, I expected them to be related, but the need to introduce a braiding suggests otherwise.

Our response:

The Fourier transform in the local Hilbert space of a single edge is similar to the change of basis in a quantum gauge theory from the group basis to a basis of eigenstates of the electric field. Nevertheless, if we want to define the total Hilbert space of the Fourier-transformed model, the braiding structure is needed to ensure that the Fourier-transformed model is topological.

The referee writes:

— Similarly, the authors should clarify the following. In the original model, violations of the vertex term are clearly given by representations of $G$, while violations of the plaquette term are given by group elements. I don't see a discussion of the excitations in the Fourier transformed model (when tails are included). This is implicit in the construction but some readers may benefit from a more detailed discussion.

Our response:

In 3D GT models with gauge group $G$, charge excitations are labeled by irreducible representations of $G$, which can be seen explicitly from the Fourier-transformed expression of the vertex operator. Nevertheless, the classification of flux excitations in the 3D GT model is very tedious when $G$ is non-Abelian (only when $G$ is Abelian violations of the plaquette term are given by group elements). A detailed discussion of excitations in these models will be referred to in future work.

The referee writes:

— I am not sure I understand Eq. 39. In Walker and Wang's formulation, the $R$ factors appear in two ways. First, in the Hamiltonian, from “twisting” an edge relative to another edge with which it shares a trivalent vertex. Second, they dictate relative phases between different edge configurations, in which the configuration effectively has a twisted loop. The picture in Eq.(58) is the one relevant to evaluating the plaquette operator. Fig.5 appears to be illustrating a situation with a crossing of the first type, in which the $R$ matrix should be telling us about a relative phase between two configurations in a ground state. Could the authors clarify the connection between this and Eq.38? It is not obvious to me.

Our response:

The two appearances of the $R$-matrix are exactly the same. Generally, a category is said to be braided if for arbitrary two objects $X$ and $Y$ we have an isomorphism $c_{X,Y}:X\otimes Y\to Y\otimes X$. As the tensor products of the group representations are commutative up to isomorphism, the category $\operatorname{Rep}(G)$ is always braided. The morphism $c_{\mu,\nu}$, where $\mu,\nu\in L_G$ then enables us to define the state with crossing edges in the graph. For example, the ket in Eq.(38) should be viewed as the morphism $c_{\mu,\nu}$. In order to evaluate the inner product in Eq.(38), we further need to define the tensor representation of the morphism, which naturally introduces the $R$-matrix as shown in Eq.(39). The $R$-matrix fully determines the isomorphism between two morphism spaces $\operatorname{Hom}(\rho,\mu\otimes\nu)$ and $\operatorname{Hom}(\rho,\nu\otimes\mu)$, as shown in Eq.(40). Eq.(40) enables us to “twist” an edge relative to another edge with which it shares a trivalent vertex and evaluate the plaquette operator.

The referee writes:

— I am also confused about how we choose $R$. The authors seem to be implying in the discussion below (41) that there is a choice of $R$. However, for general $G$, I expected there to be multiple choices. First, naively I expected that taking $R$ to be trivial (always +1) is always allowed, and that this is the choice that would have the same spectrum as the original $G$ theory (i.e. bulk point-like charges and line-like vortex loops). Second, I would also have expected that there are other choices of $R$ that satisfy the hexagon equations, and that these correspond to physically different theories in which the number of point particles in the bulk is reduced. For example, when the Hilbert space on each edge is just $\operatorname{Rep}(\mathbb Z_2 ) = \operatorname{Vec}(\mathbb Z_2)$, there are two choices for $R$, and these yield the 3D Toric code (with point particles and vortices) and the “3D semion model” which does not have point-like bulk excitations. The authors should address this point more thoroughly in Section 3, so that readers can understand how much freedom there is in choosing $R$, and how the choice of R affects the final theory. In my view this is a very central question, since it determines whether we are discussing “the” Fourier transform of the model in question, or simply “a” Fourier transform…

Our response:

We are very grateful that you have pointed out the confusion of how to choose $R$, which is indeed important and unclarfied in our paper. In fact, for any $R$-matrix which satisfied those conditions discussed in Section 3, the Fourier transform can be well defined, and hence we will get a series of Fourier-transformed models differed by the choices of the $R$-matrix. Nevertheless, all of these models are physically equivalent. We have added some discussion on the choices of the $R$-matrix in Section 3. In Section 5 we also mention that WW models with the same $F$-symbol and different $R$-matrices are also equivalent.

3D toric code model and 3D semion model are actually differed by $F$-symbols. The 3D toric code model is equivalent to the 3D GT model with input data $\mathbb Z_2$, while the 3D semion model is equivalent to the 3D twisted GT model with input data $\mathbb Z_2$ and the non-trivial 4-cocycle $\omega\in H^4(\mathbb Z_2,U(1))$. As the Fourier transforms of cocycles are really hard, in this paper we do not discuss twisted GT models or twisted boundary.

The referee writes:

— Please discuss the role of $R$ at the boundary in more detail. Again, comparing the 3D Toric code to 3D semion model, we see that the former admits two distinct gapped boundaries (which are usually thought of as flux condensing and charge condensing), while the latter apparently only admits one. Hence I would have thought that there are some stipulations on what boundary condition(s) can be chosen, and that these depend on $R$.

Our response:

As explained earlier, given bulk input data $G$ and boundary input data $K$, we have a series boundary theory of the Fourier-transformed models differed by the choices of the $R$-matrix. Similarly, these boundaries are physically equivalent.

The referee writes:

Section 4:
— In the third paragraph of Section 4, the authors should specify whether they are discussing 2+1 D or 3+1 D topological orders. They should also clarify the relationship between this paragraph and the discussion of coupled layer constructions in the following paragraph.

Our response:

From the Fourier transform introduced in this paper, we can see that, in three-dimensional topological order, the charge condensation at the boundary is completely described by the boundary condition. This statement can also be understood through the layer construction, and thus we add a paragraph to discuss it. To avoid confusion, we have moved the discussion of layer construction to the end of the section and added an explanation.

The referee writes:

— Loop-like excitations in 3D abelian topological orders are, however, well understood. The authors should note this and include Ref. https://journals.aps.org/prb/abstract/10.1103/PhysRevB.91.165119.

Our response:

We appreciate your reference to this article. We have therefore made changes to reflect the study of loop-like excitations in this article. Nevertheless, they only studied the braiding statistics of particle-like and loop-like excitations in 3D gauge theories with finite, Abelian gauge group, while generally the input data of the 3D GT model discussed in our paper could be non-Abelian. As far as I know, the classification of loop-like excitations in 3D gauge theories with finite non-Abelian gauge groups is very complicated (see Ref. 48-49 in the revised version). Describing these excitations is beyond the scope of this work. Moreover, loop-like excitations in the Fourier-transformed model, i.e., the WW model, are much less well understood. We believe that the Fourier transform constructed in this work will be helpful in studying loop-like excitations in the WW model.

The referee writes:

— The authors should add a discussion of the physics of these gapped boundaries. For example, given a subset of charges that can condense, what happens to loops that are brought to the boundary?

Our response:

Loop condensation on the boundary of the 3D GT model with a finite Abelian gauge group is obvious and has already been discussed in many publications. Nevertheless, study loop condensation on the boundary of the 3D GT model with a finite non-Abelian gauge group is extremely complicated. Therefore, these results will be reported in future work.

The referee writes:

Section 5:
— I don't understand what the authors mean when they say: “no loss of information is “the WW model does not contain charge and dyon excitations, just like the LW model.” Dyons in 3D theories would require point-like monopoles carrying charge — but the discrete gauge theories discussed here don't have magnetic monopoles. So why are we talking about dyons at all?

Our response:

We appologize for the confusion here. In fact, we just want to explain that the Hilbert space $\tilde{\mathcal H}_0^\text{GT}$ can be identified with the Hilbert space of the WW model with no charge excitations, which is the Hilbert space where the plaquette operators of the WW model are actually defined. To make the discussion here more accurate, we have made corresponding changes in the article.

We are grateful for the opportunity to improve our manuscript and are willing to make additional changes if necessary. We hope that our responses and revisions will address the concerns raised.

---

## Round 2 · Referee Report · Anonymous (Referee 2) · 2025-2-9

Strengths

The authors have made some improvements to clarity, and to the possible applications of this formalism.

Weaknesses

The paper is still extremely technical and of interest to a fairly narrow audience. The authors could do more (see requested changes below) to both situate this work in the general literature on lattice gauge theories, and to clarify the physics of the models that they are presenting.

Report

The authors have addressed some of my comments. In particular, the clarifications about the R matrices are helpful, and the introduction has been clarified.

However, I do not feel that the authors have addressed all of the recommended changes, and some things still require attention. Of particular importance are those comments aimed at increasing the general interest of the article.

Requested changes

— I feel strongly that the authors should address in the text the connection between what they are doing, and non-abelian lattice gauge theory written in the electric/ magnetic bases, as described in classic work by e.g. Kogut. I recommend adding a couple of sentences in the introduction about the relationship between the Fourier transform and this change of basis at the Hamiltonian level, and then adding a more detailed discussion in section 3. Situating this work in the context of the large historical literature on lattice gauge theories will make it accessible to a much wider audience.

— Something I didn’t notice last time: In the discussion of gapped boundaries circa Eq. 5, it is not clear that the gapped boundaries introduced are a complete set, or simply some of the gapped boundaries. The authors should clarify this point. Certainly in 2 dimensions there is also a choice of cocycle involved in picking a gapped boundary.

— The authors should give more detail in terms of what they mean when they say that they believe that the elementary excitations in 3DTO’s are not fully understood at the end of section 2. Please add some specific open questions here that are not resolved in the literature. There has certainly been recent progress on defects in 3DTO’s, but I would not normally call these excitations.

— I do not feel the authors have clarified Eq. 39. When comparing to the work of Walker and Wang, it is not clear to me which edges acquire R symbols. Is the statement that one picks a projection of the 3D lattice onto 2D, and that any edges that cross in this projection will pick up an R matrix when evaluating a particular state?

— A physical interpretation for the meaning of R must be given — I find this point very confusing in the present version! If all of the models are equivalent, is the choice of R essentially a gauge choice? If so, why is it not possible to simply choose R to be trivial? For abelian gauge theories, it would seem that I can always make this choice. Is there something different in the non-abelian case? This would potentially be a place where discussing examples would be helpful. The authors also are claiming, without proof, that these models are physically equivalent; they should at least indicate at the end of section 3.3 what the argument is (or where in the text it can be found).

— In discussing the boundary, it is natural to ask whether R has to be the same at the boundary as in the bulk, or whether there is some freedom. Can the authors clarify this?

— The authors should add to the text some discussion of the excitations in the Fourier transformed model. If providing a general construction of the membrane operators that create vortex loops, and the string operators creating charges, is beyond the scope of the present paper, at least a basic discussion including where the different kinds of excitations are found (plaquettes, vertices, tails) and which ones correspond to charges/ which to fluxes in the gauge theory should be added.

— The authors should clarify the role of the tails in the Hilbert space. In a Walker-Wang model (or generalized 3D Toric code), such tails are not needed to capture all of the excitations of the discrete gauge theories.

— My previous question about the fate of loops at the boundaries is not a request to investigate other boundary types. Rather, I simply think the authors should clarify, for the boundaries that are already described in their paper, what happens to various bulk excitations when they come to the boundary. I presume that loops can break open, but the end-points remain linearly confined, but this is not discussed at all in the text.

— The authors should add at least one example — perhaps a dihedral group. A concrete example would make it much easier for readers to understand how the actual data is obtained, and also what the potentially new boundary conditions are. The paper’s main application is supposed to be to construct new types of boundaries, so an example of concretely which boundaries it realizes would be extremely helpful.

Recommendation

Ask for major revision

  • validity: high
  • significance: ok
  • originality: ok
  • clarity: good
  • formatting: excellent
  • grammar: good

Author:  Siyuan Wang  on 2025-03-18  [id 5297]

(in reply to Report 1 on 2025-02-09)

We would like to express our sincere gratitude to the editor and the referee for their continued time and effort in reviewing our manuscript. Below are our responses to the referee's questions.

The referee writes:

— I feel strongly that the authors should address in the text the connection between what they are doing, and non-abelian lattice gauge theory written in the electric/magnetic bases, as described in classic work by e.g. Kogut. I recommend adding a couple of sentences in the introduction about the relationship between the Fourier transform and this change of basis at the Hamiltonian level, and then adding a more detailed discussion in section 3. Situating this work in the context of the large historical literature on lattice gauge theories will make it accessible to a much wider audience.

Our response:

The required discussion is added. See the changing list for detail.

The referee writes:

— Something I didn’t notice last time: In the discussion of gapped boundaries circa Eq. 5, it is not clear that the gapped boundaries introduced are a complete set, or simply some of the gapped boundaries. The authors should clarify this point. Certainly in 2 dimensions there is also a choice of cocycle involved in picking a gapped boundary.

Our response:

Generally, for a GT model with input data $G$, the gapped boundary is specified by a subgroup $K\subseteq G$ and a 3-cocycle $\alpha\in H^3[G, U(1)]$, which is mentioned in the introduction. Therefore, there do exists a choice of 3-cocycle involved in picking a gapped boundary. Nevertheless, in our paper we do not discuss any twist, and hence only some of the gapped boundaries, that is, those boundaries with trivial $\alpha$, are discussed in our paper, which has been clarified in the revised version.

The referee writes:

— The authors should give more detail in terms of what they mean when they say that they believe that the elementary excitations in 3DTO’s are not fully understood at the end of section 2. Please add some specific open questions here that are not resolved in the literature. There has certainly been recent progress on defects in 3DTO’s, but I would not normally call these excitations.

Our response:

For example, a general construction for the membrane operators generating loop excitations in 3DTO's is still unkown. Moreover, our understanding of nontrivial braidings of loops (three-loop braiding) in 3DTO's is limited to the abelian case (see for example, https://doi.org/10.1103/PhysRevB.99.235137). Nevertheless, in order to avoid dispute, we have already modified our discussion in the revised version.

The referee writes:

— I do not feel the authors have clarified Eq. 39. When comparing to the work of Walker and Wang, it is not clear to me which edges acquire $R$ symbols. Is the statement that one picks a projection of the 3D lattice onto 2D, and that any edges that cross in this projection will pick up an $R$ matrix when evaluating a particular state?

Our response:

Yes. During the Fourier transform, firstly one picks a projection of the 3D lattice onto 2D, and any two edges that cross in this projection will pick up an $R$ matrix when evaluating a particular state, i.e., evaluating the inner product Eq.(38), which has been clarified in the revised version.

The referee writes:

— A physical interpretation for the meaning of R must be given — I find this point very confusing in the present version! If all of the models are equivalent, is the choice of R essentially a gauge choice? If so, why is it not possible to simply choose R to be trivial? For abelian gauge theories, it would seem that I can always make this choice. Is there something different in the non-abelian case? This would potentially be a place where discussing examples would be helpful. The authors also are claiming, without proof, that these models are physically equivalent; they should at least indicate at the end of section 3.3 what the argument is (or where in the text it can be found).

Our response:

Discussing the physical meaning of $R$ is beyond the scope of our work. We just find that in order to define the Fourier transform of the 3D GT model, one have to fix a set of $R$-symbols. For the abelian cases, we can always choose $R$-symbols to be trivial. Nevertheless, when the input data $G$ is non-abelian, then in most cases $R$-symbols cannot be chosen to be trivial. The simplest example is the dihedral group $G=D_3$. Appendix C is added to list the data for $G=D_3$.

We also state that models with different $R$-symbols are physically equivalent. Although it is hard to write down a rigrous proof, we have the argument as follows:

Starting with a GT model with input data $G$ (denoted by $\text{GT}_G$), given the UFC $\text{Rep}(G)$ equipped with a set of $R$-symbols, we can define a Fourier transform, which maps the Hamiltonian of the GT model to the Hamiltonian of a WW model term by term. This transformation (denoted by $\text{FT}_R$) is a unitary linear transformation that does not affect the spectrum of the Hamiltonian and the ground state degeneracy. Therefore, the resulting WW model (denoted by $\text{WW}_{\text{Rep}(G),R}$) is physically equivalent to the original GT model $\text{GT}_G$. Nevertheless, if we choose another set of $R$-symbols denoted by $R'$, we will get another WW model $\text{WW}_{\text{Rep}(G),R'}$ via the transformation $\text{FT}_{R'}$, which is also physically equivalent to the original GT model $\text{GT}_G$. Therefore, the two WW models $\text{WW}_{\text{Rep}(G),R}$ and $\text{WW}_{\text{Rep}(G),R'}$ are related through the transformation $\text{FT}_{R'}\circ\text{FT}_R^{-1}$, and hence must be physically equivalent. The argument above is added in the revised version.

The referee writes:

— In discussing the boundary, it is natural to ask whether $R$ has to be the same at the boundary as in the bulk, or whether there is some freedom. Can the authors clarify this?

Our response:

By our construction, it is natural to choose $R$ to be the same at the boundary as in the bulk, because boundary edges may have crossing with bulk edges. Although there may be some special cases where one can choose a different set of $R$-symbols than in the bulk, these cases are beyond our discussion.

The referee writes:

— The authors should add to the text some discussion of the excitations in the Fourier transformed model. If providing a general construction of the membrane operators that create vortex loops, and the string operators creating charges, is beyond the scope of the present paper, at least a basic discussion including where the different kinds of excitations are found (plaquettes, vertices, tails) and which ones correspond to charges/ which to fluxes in the gauge theory should be added.

Our response:

The required discussion for charges is added. See the changing list for detail. In Section 5.2, we have already discussed where flux excitations (i.e., the loop-like excitations) are found.

The referee writes:

— The authors should clarify the role of the tails in the Hilbert space. In a Walker-Wang model (or generalized 3D Toric code), such tails are not needed to capture all of the excitations of the discrete gauge theories.

Our response:

In fact, to capture all of the charge excitations in the WW model, those tails are needed, which is analogous to the tails in the extended LW model discussed in http://dx.doi.org/10.1103/PhysRevB.97.195154. This reference is added in the revised version.

The referee writes:

— My previous question about the fate of loops at the boundaries is not a request to investigate other boundary types. Rather, I simply think the authors should clarify, for the boundaries that are already described in their paper, what happens to various bulk excitations when they come to the boundary. I presume that loops can break open, but the end-points remain linearly confined, but this is not discussed at all in the text.

Our response:

Physically, we do agree that loops can break open with linearly confined end points at the boundary. Nevertheless, to study these string excitations with two end points attaching to the boundary concretely, we still need to construct the membrane operator which generates such excitations, which is beyond the scope of our work.

The referee writes:

— The authors should add at least one example — perhaps a dihedral group. A concrete example would make it much easier for readers to understand how the actual data is obtained, and also what the potentially new boundary conditions are. The paper’s main application is supposed to be to construct new types of boundaries, so an example of concretely which boundaries it realizes would be extremely helpful.

Our response:

Two examples: abelian group $\mathbb Z_n$ and dihedral group $D_3$ are discussed in Appendix C.

We are grateful for the opportunity to improve our manuscript again and are willing to make additional changes if necessary. We hope that our responses and revisions will address the concerns raised.

---

## Round 2 · Author Response

Thank the referee for taking the time to review our manuscript and for providing valuable feedback. We are grateful for the insightful comments and the opportunity to address them. Based on the referee's report feedback, we refined our manuscript. Notably, we have expanded the discussion on the choices of the R-matrix and clarified our definition of the Fourier transformation. We hope that these revisions address the referee's concerns and make our manuscript suitable for publication in SciPost Physics. We look forward to further feedback on our revised manuscript.

---

## Round 2 · List of Changes

1. On page 3, the third paragraph of subsection 1.2, a sentence is added to explain the phrase ``UBFC $\mathrm{Rep}(G)$'', as in general literature the category $\mathrm{Rep}(G)$ is not required to be braided.

2.On page 6, the first paragraph of section 3 is completely rewritten in order to emphasize that the Fourier transform introduced in section 3 is determined by the data ${C_{\mu\nu\rho},R^\rho_{\mu\nu}}$.

3.On page 16, at the end of subsection 3.2, a discussion on the choices of the $R$-matrix is added.

4.On page 21, the fourth paragraph of subsection 4.2, we revised our discussion of the classification of loop-like excitations in 3D topological orders to include recent progress. References 47-49 are also added.

5.On page 21, subsection 4.2, we have moved the discussion of layer construction to the last paragraph and added an explanation.

6.On page 22, subsection 5.1 is rewritten to explain that the Hilbert space $\tilde{\mathcal H}_0^\text{GT}$ can be identified with the Hilbert space of the WW model with no charge excitations, which is the Hilbert space where the plaquette operators of the WW model are actually defined. The confusing discussion about dyon is removed.

  1. On page 23, below Eq.(59), a sentence is added to explain the necessity of the braiding structure in evaluating the Fourier-transformed plaquette operator.

8.On pages 24-25, the last paragraph of subsection 5.2, we add the conclusion that the three-dimensional GT model with input data $G$ is equivalent to the WW model with input data $\mathrm{Rep}(G)$, and different WW models, each with input data $\mathrm{Rep}(G)$ but equipped with distinct braiding structures, are also equivalent.

---

## Round 3 · Author Response

Thank you for your continued time and effort in reviewing our manuscript. We greatly appreciate your insightful feedback, which has helped us further improve our work. In response to your latest comments, we have focused on incorporating a specific example to better illustrate our key arguments. We believe this addition clarifies our discussion and strengthens the presentation of our results. We hope these revisions address your concerns and bring our manuscript closer to meeting the standards for publication in SciPost Physics. We look forward to any further feedback you may have.

---

## Round 3 · List of Changes

1. On page 4, a footnote is added, where we have mentioned the relationship between the Fourier transform and this change of basis at the Hamiltonian level of non-abelian lattice gauge theory.

  2. On page 4, the fourth paragraph of subsection 1.2, a sentence is added to further explain the role of the tails in the Hilbert space. Reference 40 is added there.

  3. On page 6, the third paragraph of section 2, a sentence is added to clarify that only some of the gapped boundaries, that is, those boundaries with trivial twist, will be discussed in our paper.

  4. On page 6, the last sentence of section 2, we change the phrase "elementary excitations" to "loop-like excitations". Reference 45, studying 3-loop braiding in 3DTO's, is also added there.

  5. On page 12, below eq.(32), we add some text to compare our Fourier transform with the basis transformation between the electric and magnetic bases of the non-abelian lattice gauge theory.

  6. On page 16, below eq.(40), a sentence is added to emphasize that any two edges that cross in this projection will pick up an $R$ matrix when evaluating a particular state.

  7. On page 16, at the end of subsection 3.2, a sentence is added to indicate that the argument for the equivalence of models with difference $R$-matrices can be found in subsection 5.2, and another sentence is added to indicate that some examples can be found in appendix C.

  8. On page 17, at the end of subsection 3.3, a paragraph is added to discuss charge excitations in the bulk.

  9. On page 26, at the end of subsection 5.2, the argument for the equivalence of models with difference $R$-matrices is added.

  10. Appendix C is added to list some examples.

---

## Editorial Decision

published